# Interactions between callose and cellulose revealed through the analysis of biopolymer mixtures

Radwa H. Abou-Saleh[1,2], Mercedes C. Hernandez-Gomez[1], Sam Amsbury [1], Candelas Paniagua[1], Matthieu Bourdon[3], Shunsuke Miyashima[4], Ykä Helariutta [3], Martin Fuller[1], Tatiana Budtova[5], Simon D. Connell [6], Michael E. Ries [7] & Yoselin Benitez-Alfonso [1]

The properties of (1,3)-β-glucans (i.e., callose) remain largely unknown despite their importance in plant development and defence. Here we use mixtures of (1,3)-β-glucan and cellulose, in ionic liquid solution and hydrogels, as proxies to understand the physico-mechanical properties of callose. We show that after callose addition the stiffness of cellulose hydrogels is reduced at a greater extent than predicted from the ideal mixing rule (i.e., the weighted average of the individual components' properties). In contrast, yield behaviour after the elastic limit is more ductile in cellulose-callose hydrogels compared with sudden failure in 100% cellulose hydrogels. The viscoelastic behaviour and the diffusion of the ions in mixed ionic liquid solutions strongly indicate interactions between the polymers. Fourier-transform infrared analysis suggests that these interactions impact cellulose organisation in hydrogels and cell walls. We conclude that polymer interactions alter the properties of callose-cellulose mixtures beyond what it is expected by ideal mixing.

[1] Centre for Plant Science, School of Biology, University of Leeds, Leeds LS2 9JT, UK. [2] Faculty of Science, Biophysics Division, Department of Physics, Mansoura University, Mansoura, Egypt. [3] The Sainsbury Laboratory, University of Cambridge, Bateman Street, Cambridge CB2 1LR, UK. [4] Graduate School of Biological Sciences, Nara Institute of Science and Technology, 8916-5 Takayama, Ikoma, Nara 630-0192, Japan. [5] MINES ParisTech, Centre for Material Forming (CEMEF), PSL Research University, UMR CNRS 7635, CS 10207, 06904 Sophia Antipolis, France. [6] Molecular and Nanoscale Physics Group, School of Physics and Astronomy, University of Leeds, Leeds LS2 9JT, UK. [7] Soft Matter Physics Research Group, School of Physics and Astronomy, University of Leeds, Leeds LS2 9JT, UK. These authors contributed equally: Radwa H. Abou-Saleh, Mercedes C. Hernandez-Gomez. Correspondence and requests for materials should be addressed to M.E.R. (email: m.e.ries@leeds.ac.uk) or to Y.B.-A. (email: y.benitez-alfonso@leeds.ac.uk)

The cell wall of plant, bacteria and fungi are enriched with β-glucans which are essential for development and adaptation to a changing environment[1,2]. The physical and mechanical properties of this diverse group of polysaccharides are applied in industry to obtain many valuable products. For example, β-glucans cell wall biopolymers are used as raw materials for paper, films and textiles, as gelling agents and natural immune stimulants in food and in pharmaceutical products[3]. They are also applied in the development of recent technologies such as tissue engineering, prosthetics and nanostructured materials[4,5]. In the cell wall matrix, the composition and structure of β-glucans underpin plant growth and development[6]. Understanding how these glycans are assembled, how they interact at the molecular level, and the specific physical and mechanical properties they confer, is a major challenge due to the complexities of their natural systems. This knowledge could open a whole new range of applications in biotechnology and in the design of new biodegradable materials.

Cellulose (a homopolymer of (1,4)-β-linked glucosyl residues) is the most abundant β-glucan polysaccharide, found in the cell wall of bacteria, algae, fungi and plants and even in urochordate animals[7]. Due to numerous intra- and intermolecular hydrogen bonds and van der Waals interactions[8] cellulose chains self-assemble into microfibrils which provide strength and determines the direction of cell growth through increasing the stiffness of transverse cell walls. Interactions between cell wall polymers modify the ordering, spacing or slippage between microfibrils affecting cell wall mechanics and thereby cell behaviour such as growth rate, shape, etc[6,9,10]. The interactions of cellulose with diverse glycan polymers (including xyloglucans and xylans) have been reported but rarely validated in biological systems[11–13]. The association of these macromolecules into a composite structure is dependent upon intermolecular bonding, hence stability depends on several factors including the polymer side-chains and the number and spatial configuration of these interactions[14–16].

In contrast to cellulose, plant (1,3)-β-glucans (termed callose) are a minor component in developing cell walls but have important regulatory functions in cell-to-cell signalling, organogenesis and defence[17,18]. In normal unstressed conditions, callose appears mainly deposited around intercellular junctions named plasmodesmata[19]. Callose accumulation determines intercellular communication, affecting developmental and environmental signalling[20]. Callose also has important regulatory functions in cell plate formation during cell division and in the sieve pores that connect the phloem vascular system[21]. It accumulates in response to wounding and exposure to biotic and abiotic stresses playing a key role in defence[22,23].

Despite its importance, the physical properties of callose and its interactions with other biopolymers within the cell wall matrix remain unknown. Most of the work in this area focuses on callose deposited during pathogenic attacks or during pollen tube growth. Callose is thought to strengthen the cell wall forming a physical barrier against fungal penetration[24–26] and to act as a matrix for the deposition of other cell wall components[27]. Digestion of callose appears to reduce cell wall stiffness leading to an increase in the diameter of the pollen tube[28,29]. Conversely, a mutant in a callose biosynthetic enzyme (named *powdery mildew resistant 4*, *pmr4*), shows reduced callose levels but appears resistant to some fungal infections[30]. Pectin and cellulose distribution also change after digestion of callose in pollen tubes[28] but the molecular mechanism behind these changes and their implications for plant-pathogen interactions are not clear[31].

Recent studies visualised a 3D network of callose-cellulose microfibrils forming in the site of fungal penetration[32]. Given the importance of callose to regulate developmental and adaptive processes it is imperative to clarify whether it can interact with cellulose at the molecular level and what are the mechano-physical properties that derive from this interaction. New emerging reagents and tools can be used in dissecting this question, including ionic liquids, such as 1-ethyl-3-methylimidazolium acetate ([C2mim][OAc]) which is a powerful solvent for cellulose and other water-insoluble polysaccharides[33], and vectors carrying a hyperactive version of the biosynthetic enzyme callose synthase 3 in an inducible system (*icals3m*)[34], which produce an over-accumulation of callose in cellulosic cell walls.

In the current study, we use a combination of techniques including: Atomic Force Microscopy (AFM) and nanoindentation analysis; Scanning Electron Microscopy (SEM), liquid state Nuclear Magnetic Resonance (NMR); rheology and Fourier Transform InfraRed (FTIR) spectroscopy to analyse the properties of mixture solutions of microcrystalline cellulose and a callose analogue (Pachyman). The results unveil molecular interactions between cellulose and callose in both the liquid solution and in hydrogels, forming systems with dynamics and mechanical properties that are not adequately described by a simple ideal mixing rule, i.e., the mixtures properties are not just the weighted average of the individual component properties. We consider the relevance of these interactions in cell walls by comparing wild type and transgenic plant material expressing *icals3m*. Based on the results, we propose that interactions occur between callose and cellulose that influence cell wall structure (and thereby properties). The implications of these results for the regulation of cell-to-cell communication and other biological processes are discussed, as well as their potential applications for the design of new cellulosic biomaterials.

## Results

**Mechanical properties of cellulose-callose hydrogels.** To gain insight into how callose modulates the physico-mechanical properties of plant cell walls, we used a simplified model of composite hydrogels, as reported elsewhere[35]. The samples were made with different proportions of commercial microcrystalline cellulose and the callose analogue Pachyman dissolved in the ionic liquid [C2mim][OAc] and coagulated in water (see details in the methods section). It should be noted that cellulose and callose are insoluble in water, and thus when water is added to their solutions in ionic liquid, phase separation occurs leading to a formation of a 3D network. We shall use the term hydrogel in the following keeping in mind that the samples were precipitated polymer networks with water in the pores. Mixtures contained a 10% total weight (wt.) of the polysaccharides with different proportions of the individual polymers. These samples are referred to as wt. percent of cellulose-callose in the mixtures. Pure cellulose is labelled 0%, 20:80 is 20% callose, 50:50 is 50% callose, 80:20 is 80% callose and pure callose is labelled as 100%. The concentration of callose in hydrogels was fluorescently quantified as described in the Methods[36] in order to confirm that callose was not washed out during exchange of ionic liquid to water. The results showed that callose was not removed during solvent exchange and its concentration in hydrogel is the same as in the mixtures with cellulose (see Supplementary Fig. 1).

The morphology of supercritically dried hydrogels was visualised using SEM (Fig. 1) to identify potential changes in their structure as a function of mixture composition. 0% callose shows the classical structure of macroporous cellulose aerogels[37]. Significant differences in morphology were detected between 0%, 50%, 80% callose and the 100% callose. The increase in callose concentration leads to a finer morphology with fewer and fewer macropores.

The mechanical properties of the hydrogels were measured using AFM nanoindentation with a colloidal probe. Force maps

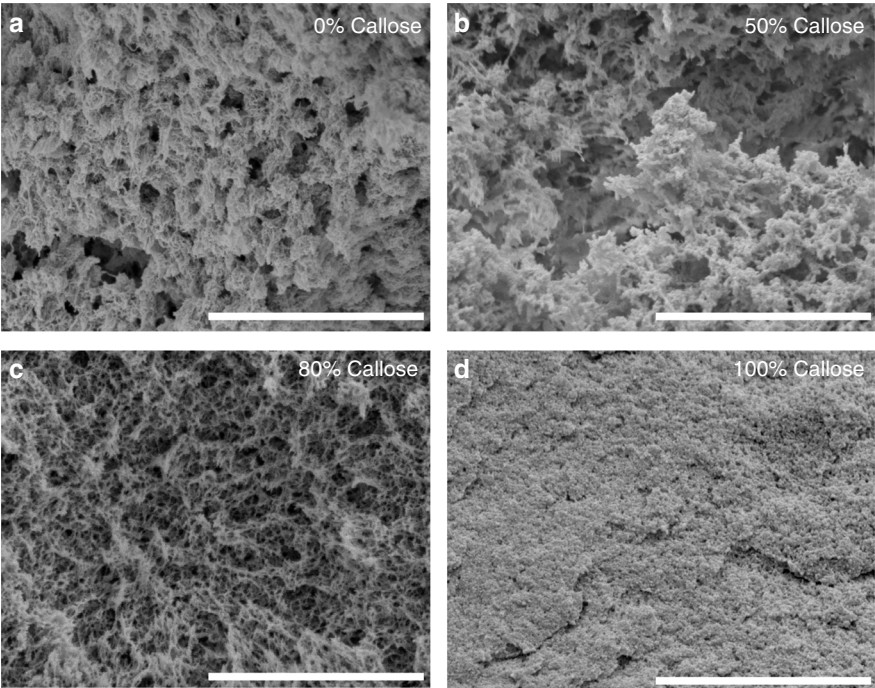

**Fig. 1** Morphology of hydrogels with different callose concentrations. Cross-sections of dried hydrogels with different amounts of cellulose and callose were prepared as described in the methods and visualised using scanning electron microscopy (SEM). The pictures (**a**–**d**) are representative of each mixture composition and are labelled as percentage of callose. Scale bar = 3 μm

were collected and force-indentation curves were fitted to the Hertz model[38,39] (Fig. 2a and Supplementary Fig. 1) to calculate the Young's modulus for the five different hydrogels. The plastic behaviour of the hydrogels (time dependent irreversible deformation in response to applied force) can be compared in a relative fashion by calculating the index of plasticity from the hysteresis, derived from the recovery of the indented surface after the force is relaxed, as described in the methods[38,40]. Mean values of Young's modulus and plasticity index are presented in Fig. 2b, c, and were derived from statistical analysis of hundreds of force curves (Supplementary Fig. 3).

The plasticity index for all samples was between 0.2 and 0.5 (where 0 represents a purely elastic response, and 1 a purely viscous or plastic response with no recovery[40,41]). Both the highest modulus value (stiffness) and plasticity index were measured for cellulose hydrogels in comparison to 100% callose. Plasticity values remained around 0.45 from 0% to 20% callose, and slowly reduced thereafter. The elastic modulus dropped dramatically from 200 ± 6.6 to 86 ± 1.6 kPa (errors are standard error of the mean) when comparing cellulose (0% callose) to 20% callose hydrogel, respectively. The modulus continues to decrease with increasing callose concentration reaching a value of 42 ± 1.6 kPa at 100% callose. The theoretical Voigt upper and Reuss lower bounds are plotted for the elastic modulus[42,43] (Fig. 2d). Although these models are typically used for composites reinforced with fibres, and in our samples there are no fibres, they mathematically describe the mechanical response of two idealised combinations of materials (not exclusive to fibres): one to maximise the stiffness by having the stiffer component spanning the sample in the direction of strain (Voigt), and one to minimise the stiffness by having the stiffer component in layers orthogonal to the strain direction (Reuss). It is therefore useful to compare our results with the limits established by these models. In the hydrogels, the experimental value of the Young's modulus corresponding to 20% callose does not fall in between these

boundaries. This indicates that the large reduction in elastic modulus upon addition of 20% callose cannot be explained by a physical arrangement of the callose and cellulose components, leading to the conclusion it must be an effect at the molecular level.

Mechanical properties of the cellulose-callose hydrogels were further explored using a 2 mm flat-ended probe in a Texture Analyser (as described in the methods) suitable for soft materials. This method allows testing beyond the elastic limit through to failure at high strain (Fig. 3). Analysis of the linear elastic region revealed an identical relationship to AFM-nanoindentation, with higher modulus at 0% callose, dropping rapidly at 10–20% callose. The absolute values are about 50% of the AFM values, which is reasonable considering the magnitudes difference in the scale of measurement, different contact geometry, contact mechanics model and boundary conditions. It is also possible that the 50% modulus reduction is a function of the indentation depth, with the 50–150 nm deformation in AFM reporting on near-surface properties, and the 2 mm flat-ended probe giving a bulk measurement with an elastic response up to around 1 mm deformation (25% strain). The largest difference between samples occurred beyond the elastic limit. Although the addition of a small amount of callose reduced the yield stress by 2–3 times, it also abolished the catastrophic failure of the pure cellulose. At the yield point (when plastic non-linear deformations begin), the pure cellulose gel suddenly cleaves to a depth of >1 mm. At 10% callose this sudden cracking is entirely abolished, and the hydrogel yields smoothly and plastically all the way through the depth, maintaining its integrity. This behaviour is similar at 20%, but the hydrogel is much weakened at 50% and greater callose concentrations, with a 20× reduction in yield stress. The uptick in force at the end of each trace is caused by the proximity of the underlying surface on which the gel sits.

In summary, mechanical testing using both AFM-nanoindentation and a Texture Analyser in compression mode

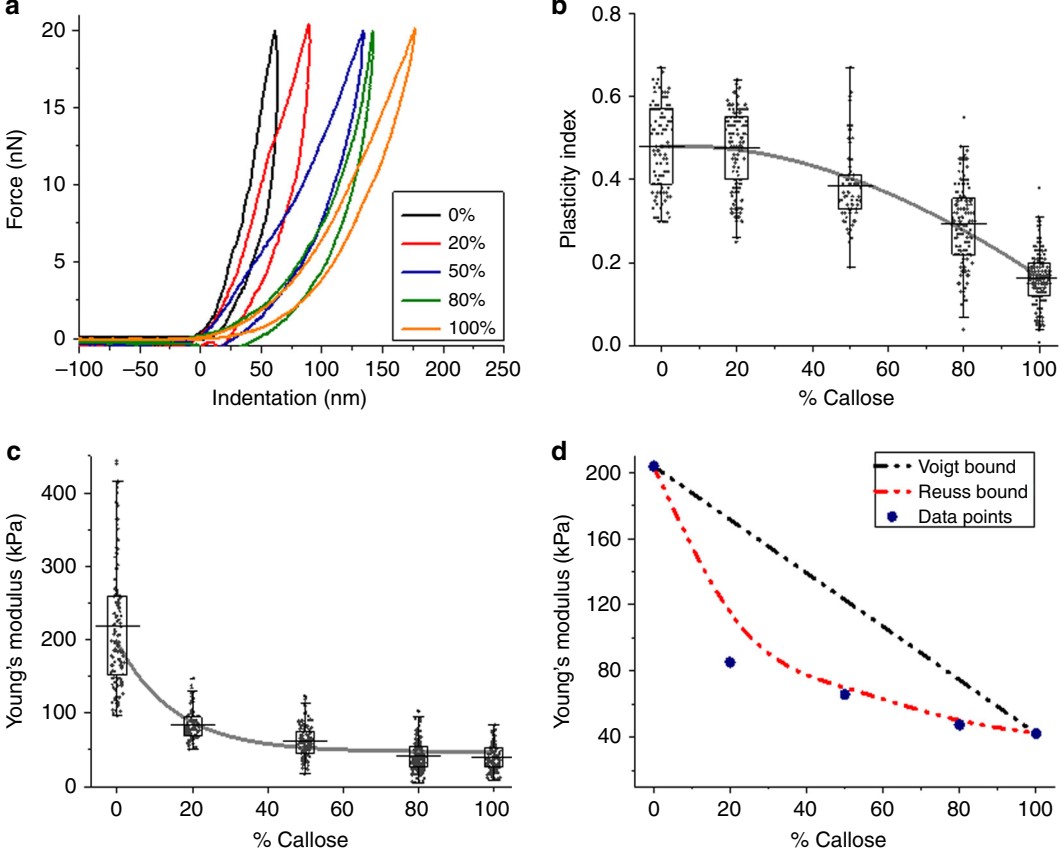

**Fig. 2** Nano-mechanical properties of the hydrogels. **a** shows typical loading and unloading force curves obtained for the different hydrogels (0% callose in black, 20% in red, 50% in blue, 80% in green an 100% in orange) using AFM-nanoindentation. **b** shows plasticity values as a function of callose concentration; the solid line is presented to guide the eye. **c** shows the calculated Young's modulus (data points) as a function of the callose concentration in the hydrogel (solid line is to guide the eye). **d** Mean Young's modulus (blue circles, individual data points displayed in **c** are represented in relation to values obtained as a result of applying the Voigt and Reuss hypothetical model (dashed lines in black and red, respectively). Box plots in **b** and **c** represent the first (25%) and third (75%) quartiles, the central horizontal line is the mean, and outliers at the 1% and 99% level are indicated by the whiskers, the data points are averages for 3 individual samples, with 5 areas imaged for each sample

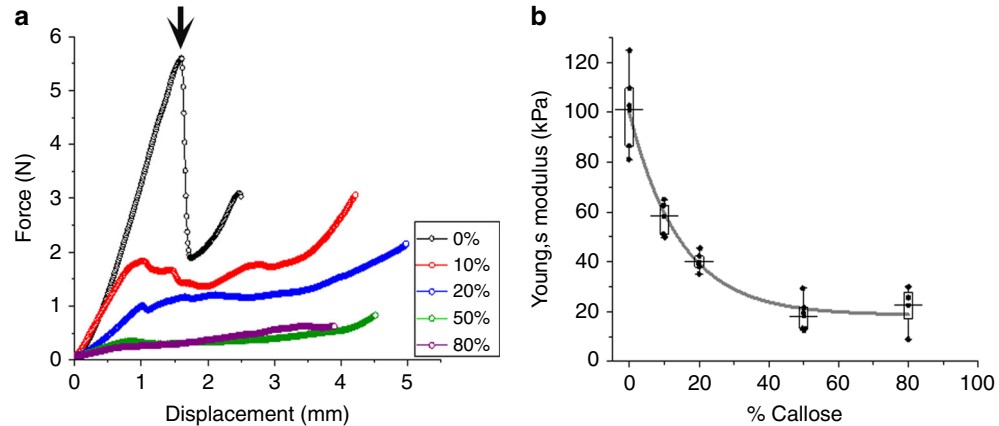

**Fig. 3** Mechanical properties of cellulose-callose hydrogels. **a** shows force-displacement curves obtained for the hydrogels at different callose concentrations (0% in black, 10% in red, 20% in blue, 50% in green and 80% in magenta) using a Texture Analyser and a 2 mm flat-ended probe in compression mode. The yield point for 100% cellulose (0% callose), which coincides with a yield stress of 1.4 MPa, is arrowed. **b** shows Young's modulus values as a function of callose concentration represented in a box plot. The solid line is drawn to guide the eye. The Young's modulus was calculated according to the Sneddon model as described in methods. Boxes represent the first (25%) and third (75%) quartiles, the central horizontal line is the mean, and outliers at the 1% and 99% level are indicated by the whiskers. Individual data correspond to four-five independent replicas

indicates that the addition of a small amount of callose strongly reduces the elastic modulus of the resultant hydrogels beyond what is predicted by the rule of mixtures (see Fig. 2d). Plasticity (and therefore the ability to absorb energy to cope with deformations) is maintained in cellulose-callose hydrogels until it drops at higher callose concentrations (Fig. 2b). The yield behaviour is also very different, with the pure cellulose gel allowing the stress to build up before the material catastrophically fails and breaks apart, whereas an addition of 10% callose allows a smooth yield and plastic deformation without a sudden failure event. The results suggest that a small amount of callose disrupts the cellulose network changing the hydrogel mechanical properties, which points to potential microscopic interactions between these two polymers. The yield behaviour might suggest that intermolecular bonding is weakened such that the polymer chains can slide more easily and relieve the build-up of stress. Hence addition of small amount of callose allows the material to be more resilient to both high stress and strain which otherwise might lead to material failure.

**Viscoelastic behaviour of cellulose-callose mixtures**. Mechanical testing of the hydrogels shows a significant drop in plasticity at concentrations >50% callose. The next step was to examine the viscoelastic properties of cellulose-callose mixtures in the liquid state. The phase angle ($\delta$) and storage ($G'$) and loss modulus ($G''$) of neat cellulose and callose solutions and of their mixtures are represented in Fig. 4; the complex modulus ($G^*$) is shown in Supplementary Fig. 4.

The data suggest gel-like properties for the neat callose solution as the phase angle remained almost independent of the frequency. This is also reflected by the overlap in the storage and loss modulus at low frequency for 100% callose (Fig. 4b).

Steady state viscosity of the mixtures was measured as a function of shear rate at different temperatures. Flow curves demonstrate mostly Newtonian behaviour and a very weak shear thinning, which is more pronounced for 100% callose solutions, likely as a consequence of high molecular weight chains' alignment in the flow (Supplementary Fig. 5). The Cross model (Eq. 1)[44] was used to fit the data and calculate the viscosity of the samples at zero shear rate:

$$\eta = \frac{\eta_0 + \eta_\infty \alpha \gamma^n}{1 + \alpha \gamma^n} \qquad (1)$$

where $\gamma$ is shear rate, $\eta_0$ is zero shear rate viscosity, $\eta_\infty$ is the viscosity at infinite shear rate, $n$ is the flow index and $\alpha$ is the consistency index.

The results show, as expected, that viscosity increased with decreasing temperature for each system studied and, consistently, with the increase of callose concentration (Fig. 5). Interestingly, at all temperatures tested, values at 80% callose show lower viscosity than predicted by a simple ideal mixture rule, as it overlaps with the ideal theoretical values for 50% callose (calculated as the average of the 0% and 100% ln $\eta_0$ values) (Fig. 5b). On the contrary, the experimental value of the zero shear rate viscosity of the mixture with 50% of callose is higher than ideal mixing rule prediction. It is clear that there are structural reorganisations in the mixtures with more than 50% of callose.

The activation energy of viscous flow was calculated using the Arrhenius approach but no significant influence of mixture composition was detected across the entire concentration range, with values between $64.2 \pm 1.8$ kJ and $70.8 \pm 9.6$ kJ.

To summarise, the rheological analysis suggests gel-like properties for 10% wt. callose-ionic liquid solution. The viscosity of the 80% callose samples coincides with the experimental and theoretical values for 50% callose suggesting that the mixtures do not obey the ideal mixing rule.

**Liquid state $^1$H-NMR suggests cellulose-callose interactions**. Molecular interactions between the two polymers could explain the non-ideal mixing behaviour in the mechanical and viscoelastic properties of the mixes. To test this hypothesis, NMR was used to determine the self-diffusion coefficients of the ions as a function of composition of mixed solutions. The ions act as probe molecules in these systems. Their diffusion coefficients depend on their environment, which can be affected by the interactions between the polymers. For cellulose, it has been shown that the ions interact with the hydroxyl groups (not engaged in forming intra or inter–molecular hydrogen bonds between the polymers) and that this interaction reduces the diffusion coefficient of the ions in solution[45]. If callose and cellulose interact, a change is expected in the number of OH groups available for the ions to interact with and thereby their diffusion coefficient to deviate from the ideal mixing rule, such as, for example, in the case of [C2mim][OAc] and water[46].

The $^1$H NMR spectra for the different mixtures were determined within a temperature range between 20 and 60 °C inclusive. The peaks corresponding to the cation and the anion were used to calculate the diffusion coefficients as described in methods[45]. Fig. 6a shows the cation and anion diffusion coefficients as a function of mixture composition at 30 °C. Predicted values for log-linear correlations (ideal mixing) were drawn based on the values obtained for 100% cellulose and 100% callose. Notice that both the diffusion coefficients of cations and anions behave similarly as a function of mixture composition and significantly deviate from the mixing rule prediction at around 20% and 80% callose. The deviations are highlighted when comparing the percentage difference between the theoretical values (calculated based on the ideal mixing rule) and the experimental values, and these changes were also found at 40 °C (Supplementary Fig. 6). Figure 6b shows the factor $D_{\text{anion}}/D_{\text{cation}} < 1$ suggesting that in these mixtures anions diffuse slower than cations despite being smaller. This is known as anomalous diffusion[47] of the ions in ionic liquid and occurs at all tested temperatures. Consistent with deviations from ideal mixing, the anomalous diffusion become more prominent at 20% and 80% callose concentrations.

The activation energy of the ions diffusing in polymer solutions and their mixtures was calculated using the Arrhenius equation. The results (Supplementary Fig. 7) show a slight reduction from $44.1 \pm 0.8$ to $41.3 \pm 0.9$ kJ mol$^{-1}$ when comparing 100% cellulose to 100% callose, respectively. This slight decrease in activation energy reflects the increase in the diffusion coefficients when comparing 100% cellulose and 100% callose.

To summarise, the results show that at certain callose concentrations in the mixture (20% and 80%) there is a deviation of both anion and cation diffusion coefficients from the expected ideal mixing rule, which is the evidence of the changes in the number/accessibility of OH groups on carbohydrate chains that interact with the ionic liquid. Since the mixtures are of the same total polymer concentration (10% total weight), these changes can only be explained by molecular interactions between the polymers, modifying, for example, the number of inter/intramolecular hydrogen bonds. The significance of this result, together with the mechanical and the viscoelastic properties of mixtures, are discussed below.

**Changes in cellulose-callose organisation exposed by FTIR**. To further understand the effect of callose interactions with cellulose in hydrogels, we used Fourier Transform InfraRed spectroscopy

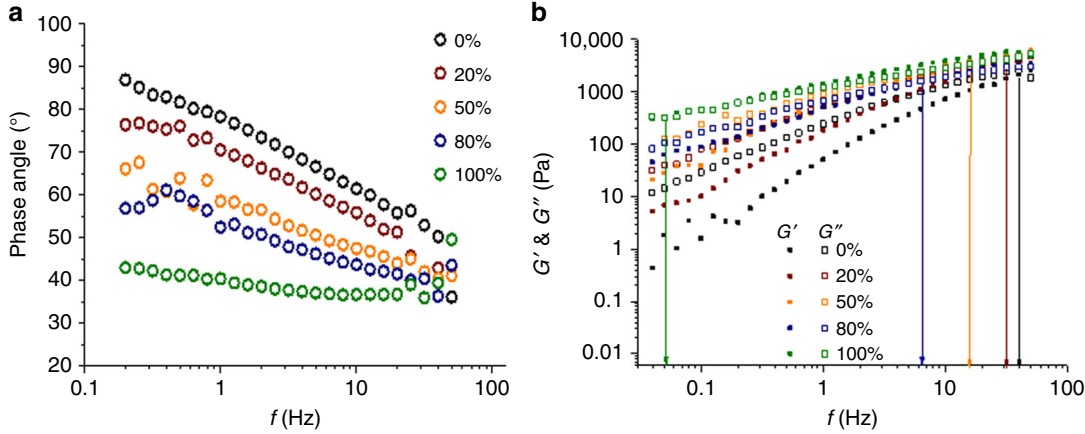

**Fig. 4** Viscoelastic properties of the cellulose-callose mixtures in ionic liquid solution at 25 °C. **a** represents changes in the phase angle ($\delta$, open circles) as a function of frequency ($f$) for the five callose concentrations. **b** Changes in the viscous ($G''$, open squares) and elastic ($G'$, filled squares) components as a function of frequency at different percentage (%) of callose in the mixtures. Graph elements in black correspond to 0% callose, in dark-red to 20%, orange to 50%, blue to 80% and green to 100%. The arrows indicate the approximate frequency where $G''$ and $G'$ overlap

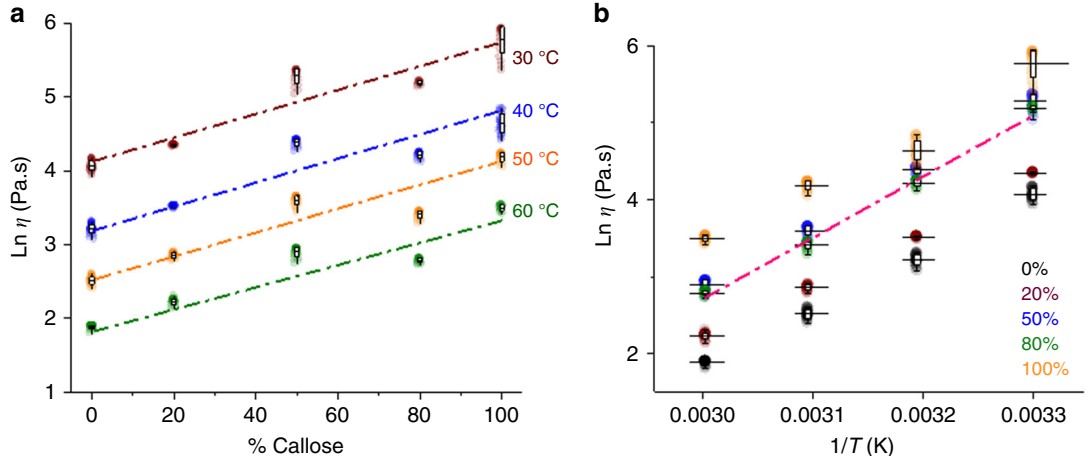

**Fig. 5** Zero shear rate viscosity values as a function of callose concentration and temperature in the mixtures of ionic liquid solution. **a** shows the Ln of the viscosity ($\eta$) at the different callose percentage and within the temperature range 30-60 °C. A linear fitting (mixing rule) is presented in discontinuous lines in brown for 30 °C, blue for 40 °C, orange for 50 °C and green for 60 °C. The graph in **b** shows changes in the viscosity values as a function of temperature at the different callose concentrations (%). The discontinuous pink line in **b** shows the theoretical values predicted for the 50% composition as calculated using the ideal mixing rule. Box plots represent the first (25%) and third (75%) quartiles, the central horizontal line is the mean, and outliers at the 1% and 99% level are indicated by the whiskers. Individual data points for four-five independent replicas are shown, with between 20 and 40 repeat measurements for each concentration/temperature combination. Every individual data point is plotted at 80% transparency

(FTIR) to analyse the spectra of the callose:cellulose mixtures (Fig. 7a). The spectra depicted main characteristic vibrational bands corresponding to cellulose II which allow characterisation of polymer mixture ordering[48]. The peak positions for the raw powders of microcrystalline cellulose (Avicel) and callose (Pachyman), as well as their 50% mixture were also obtained (Supplementary Fig. 8, Supplementary Table 1).

Significant differences were observed in the position of the OH- stretching vibration broad band (3100–3600 cm$^{-1}$) that informs about the hydrogen bonding pattern[48,49]. Using information from the spectra, the lateral order index (LOI, ratio between CH2 bending and C–O–C stretching absorption bands) and the energy of the hydrogen bond ($E_H$) was calculated (see methods and reference[49]) (Fig. 7b, c). It was found that the cellulose sample has the highest LOI (also known as crystallinity index) and $E_H$ values which correlates with forming a higher organised network in relation to callose. Both values decreased with the increase of callose concentration in hydrogel but

deviations from an ideal behaviour (linear correlation with concentration, Fig. 7b) were significant at 20% callose. This suggests that addition of callose modifies cellulose structuring during coagulation in a manner that affects the hydrogen bonding and the ordering of the polymer chains in the hydrogel. This result supports the interpretation of the results obtained by NMR.

**Callose synthesis in _A. thaliana_ modifies the FTIR profile.** Callose co-exists with cellulose in specific plant cell wall domains and cell types. To determine if callose concentration _in planta_ modifies cellulose organisation (as found in hydrogels) we used a transgenic tool to induce ectopic callose synthesis in seedlings of _Arabidopsis thaliana_. A vector containing an estradiol-dependent promoter (_pG1090:XVE_) and a mutated hyperactive version of the callose synthetic enzyme CALS3 (_cals3m_) was introduced into _Arabidopsis thaliana_ to produce a stable transgenic line named

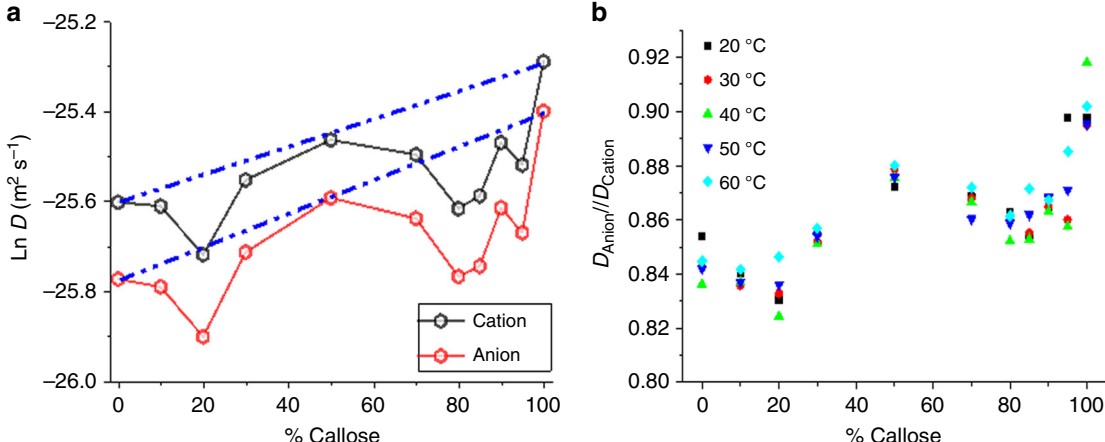

**Fig. 6** Illustration of the non-ideal behaviour of the diffusion of the ions determined by $^1H$ NMR. **a** represents the Ln diffusion coefficient ($D$) of the anion (red circles) and the cation (black circles) in [C2mim][OAc] at 30 °C as a function of mixture composition (% of callose). The blue dashed lines represent the theoretical values calculated based on the ideal mixing rule. Average percentage errors (calculated based on the uncertainty in the diffusion values for the cation) are below 2%. **b** shows the ratio of diffusion coefficients for the anion and the cation ($D_{anion}/D_{cation}$) as a function of callose concentrations at 20 °C (black square), 30 °C (red circle), 40 °C (green triangle), 50 °C (blue inverted triangle) and 60 °C (light blue diamond). The values are individual data points. Percentage differences in technical replicas at 30 °C and 40 °C are shown in Supplementary Fig. 6

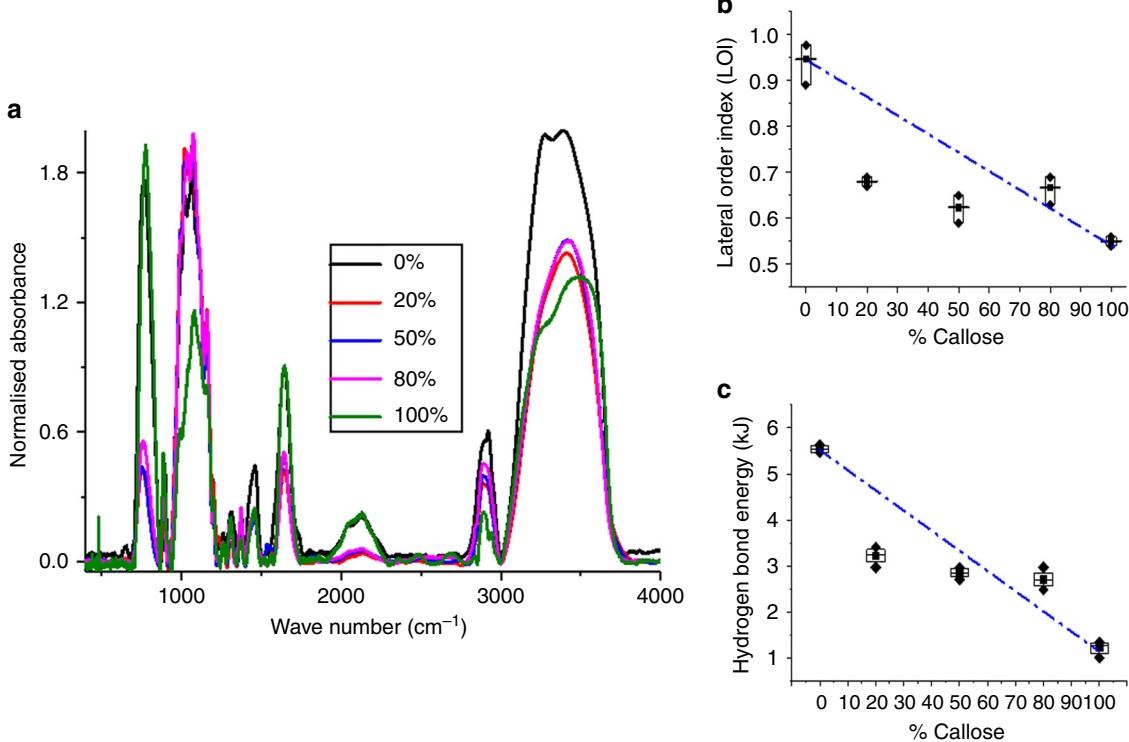

**Fig. 7** Fourier-transform infrared spectra show the spectra in hydrogels with increasing callose concentration. **a** The image shows the superimposed normalised FTIR spectra obtained for hydrogel samples with 0% (black trace), 20% (red), 50% (blue), 80% (magenta) and 100% (green) concentrations of callose. **b** shows the lateral order index (LOI) and **c** shows hydrogen bond energy as a function of the different callose concentration. These parameters measure the degree of structural organisation. In both cases, values at 20% callose deviate from ideal mixing (blue dashed line). Boxes represent the first (25%) and third (75%) quartiles, the central line is the mean, and outliers at the 1% and 99% level are indicated by the whiskers. Individual data points correspond to three independent replicas

*pG1090::icals3m*[34,50]. Phenotypic comparison after 24 h treatment of *pG1090::icals3m* with either the chemical inducer estradiol or with DMSO (control experiment) revealed a substantial increase in callose concentration in different plant tissues (detected using confocal fluorescent microscopy after aniline blue staining, Fig. 8 and Supplementary Fig. 9). Callose was quantified in cell walls isolated from 24-h-estradiol treated *pG1090::icals3m* seedlings. The amount of callose was significantly higher than wild type after icals3m induction (Table 1). Callose mainly accumulated on the vasculature and associated cells, as well as in the elongation and differentiation zones of the root (Supplementary Fig. 9). Confocal microscopy of cells at the root cap regions, co-stained

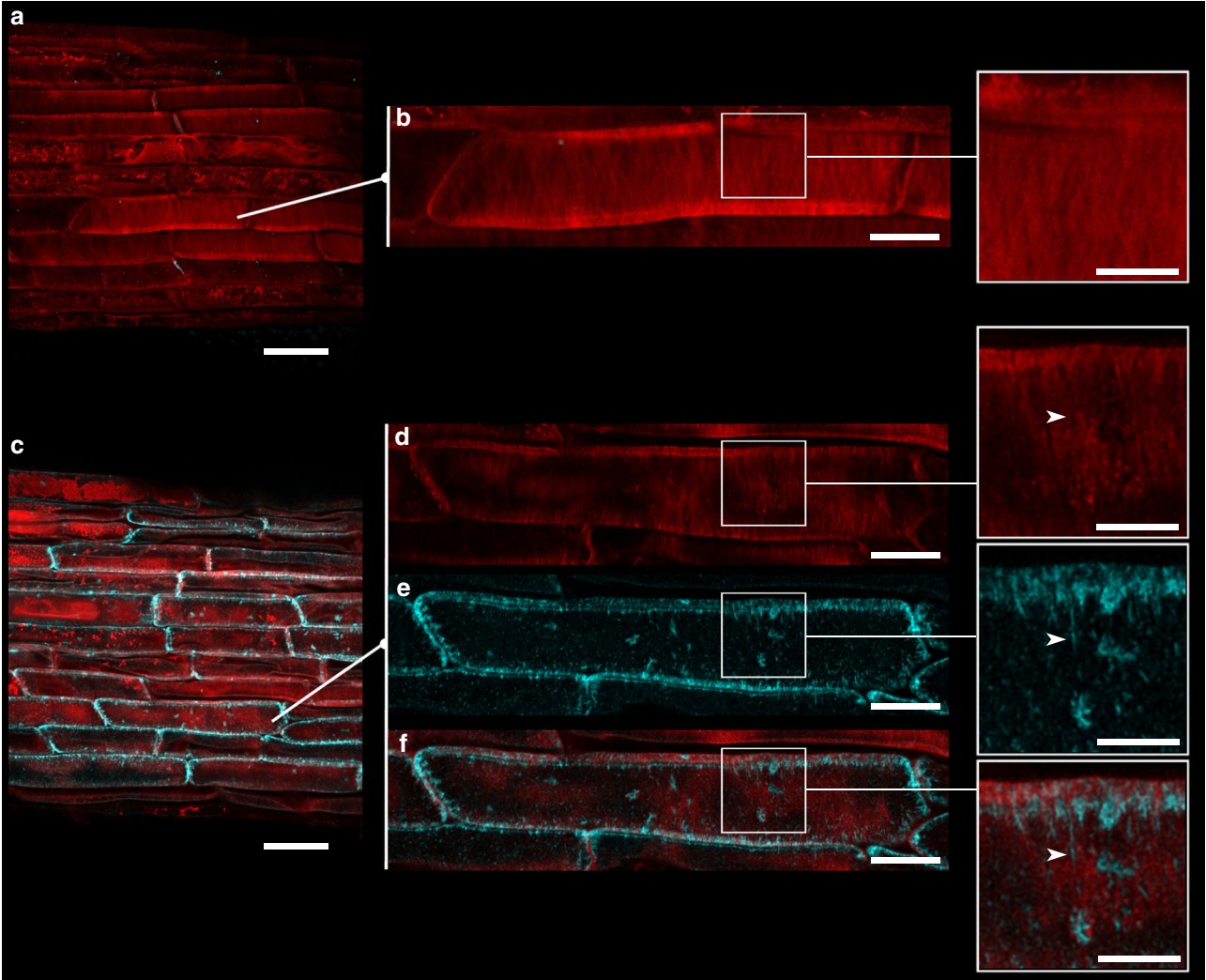

**Fig. 8** Induction of *pG1090::icals3m* increases callose which decorates cellulose microfibrils. DMSO-treated and estradiol-treated *icals3m* roots were co-stained with aniline blue (cyan) and direct red 23 (red) which fluorescently stain callose and cellulose, respectively. **a**, **b** show confocal images of a region in the elongation zone of the root 24 h after treatment with DMSO (control). **b** and close-up show the merged signal at higher magnification. **c–f** show comparative regions in *icals3m* roots 24 h after estradiol induction (activation of *icals3m*). Notice the increased accumulation of callose (cyan signal) in **c** comparing to **a**. Consult also Supplementary Fig. 9. **d**, **e** and **f** show enlargements of a cell from **c**, with their associated close-ups in the separate channels (**d**, direct red 23); **e**, aniline blue and **f**, merged image). Arrowhead highlights an area where callose integrates very closely to cellulose microfibrils. Scale bars (**a**, **b**) = 25 µm; (**b**, **d**, **e**, **f**) = 10 µm; (**b**, **d**, **e**, **f** close-ups) = 5 µm

**Table 1 Callose concentration, hydrogen bond energy and lateral order index quantified in alcohol insoluble residues (AIR) from wild type and transgenic *pG1090::icals3m* plants**

|  | Wild type[a] | *pG1090::icals3m*[a] |
| --- | --- | --- |
| Callose content ($10^{-3}$ mg per mg AIR) | 6 ± 0.72 | 11 ± 0.91 |
| Hydrogen bond energy (kJ) | 4.05 ± 0.1 | 3.38 ± 0.2 |
| Lateral order index (LOI) | 6.12 ± 0.19 | 3.85 ± 0.77 |

[a]Mean values ± standard deviation calculated for four biological replicas

strong restriction in root length as soon as 48 h after estradiol treatment/induction of callose.

FTIR was also carried out to reveal the structural organisation of cell walls after callose induction (Fig. 9). The spectra revealed peaks that coincide in position to those found in the microcrystalline cellulose (Avicel) and Pachyman (Supplementary Fig. 8). The energy of the hydrogen bonds and the LOI were quantified and significant reductions were observed after callose induction in the *pG1090::icals3m* when compared to wild type control (Table 1). The results further support the conclusion that an increase in callose content decreases hydrogen bonding coincident with more disordered cell walls. This effect might be linked to the restriction observed on root growth as discussed below.

## Discussion

(1,3)-β-glucans (i.e., callose) plays an important role in cell biology impacting organism development and environmental responses[1,51,52]. The accumulation of this polymer is associated with changes in cell wall architecture and mechanical properties

with aniline blue and Direct Red 23 (a fluorescent dye that enables visualisation of cellulose[9]), showed a pattern that suggest integration and/or close spatial proximity of callose deposits and cellulose fibrils where interactions can potentially occur (Fig. 8). Callose and cellulose concentrations and organisation in cell walls are shown to affect cell growth. Supplementary Fig. 10 shows a

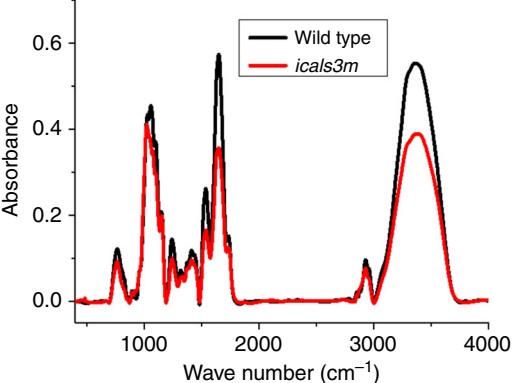

**Fig. 9** Fourier-transform infrared analysis of cell walls 24 h after estradiol treatment. The image shows an example of the FTIR profile for one replica per each genotype: wild type (black trace) and transgenic line *pG1090::icals3m* (*icals3m*, in red). Estradiol-treated *icals3m* accumulates callose in comparison to control (Fig. 8 and Supplementary Fig. 9). Notice that the absorbance of the OH- stretching vibration broad band (3100–3600 cm$^{-1}$) is reduced in *icals3m*

but it is not clear if the effect is direct or indirect. There is evidence pointing to callose functioning as a sealant, as a load-bearing structure or as a matrix for the deposition of other cell wall components[27,29,32]. Our study, using a hydrogel model, shows that the addition of callose to cellulose leads to a dramatic reduction in elastic modulus and changes the yield behaviour. Callose acts as a sort of cellulose plasticizer abolishing the dramatic failure which would otherwise occur when stress increases to the yield point and the material cleaves deeply. Rheology analysis of mixtures in ionic liquid supports a gel-like behaviour (viscoelastic response) for concentrated callose solutions. We also demonstrate that the mechano-physical properties (elastic modulus, plasticity and viscosity) of the hydrogels and mixtures in ionic liquid diverged from linear ideal mixing rules. The diffusion of the ions, measured using liquid state $^{1}$H NMR, strongly suggests that callose and cellulose engage in molecular interactions showing a non-ideal decreasing pattern at concentrations of 20% and between 70–90% callose. Deviations from ideal mixing in the hydrogen bond pattern and lateral order index were also found in 20% callose hydrogel mixtures using FTIR. Although more complex in interpretation, changes in cell walls ordering and hydrogen bond pattern also occur after increasing callose biosynthesis in plant cell walls using as a tool a transgenic estradiol-inducible callose synthase system (*pG1090::icals3m*). Together, the results indicate that the accumulation of callose causes disruption in the organisation of cellulose and cell wall networks which can influence cell wall mechanical properties and is likely linked to cell growth.

Molecular interactions between callose and cellulose are supported by the non-ideal patterns in ionic liquid across all temperature measurements recorded in the mixtures using NMR. The characteristic anomalous diffusion ($D_{anion}/D_{cation} < 1$) reported for cellulose[45] becomes more prominent at the concentrations where the diffusion of the ions follows a non-ideal pattern. The anion interacts predominantly with the carbohydrate and thus is expected to be more sensitive to the polymer-polymer interactions[47]. A decrease of $D_{anion}/D_{cation}$ indicates that the diffusion of the anion is preferentially reduced providing a strong evidence for interactions. The non-ideal behaviours of either the diffusion coefficient of the ions and of the mechanical properties of the mixtures are concentration dependent, which likely reflect the complexity of these systems where cellulose-cellulose, cellulose-callose and callose-callose interactions can contribute

differently to the molecular environment and properties of the mixtures. The fact that interactions appear to occur at certain stoichiometric ratios has been previously reported for other blends[46,53,54]. In callose-cellulose mixtures, interactions at 20% and 80% wt. appear to deviate more strongly for the diffusion coefficient of the ions in ionic liquid mixtures, i.e., more OH groups are available for the ions to interact with. The outcome of these interactions is also observed in the FTIR results where the hydrogen bond energy and lateral order index (which relate to the OH groups engaged in H-bonds) is significantly decreased in 20% callose hydrogels in relation to what was expected by the mere changes in polymer concentrations.

The biological importance of the findings using hydrogels as a model is still a point of debate. In our study, we showed that increasing callose concentration in developing cell walls of the model plant *Arabidopsis thaliana* significantly affects the hydrogen bond energy and lateral order index (considered as crystallinity index by other authors)[55] calculated using FTIR spectra of cell walls (Fig. 9, Table 1). Moreover, confocal imaging after induction of callose using the *pG1090::icals3m* line shows close proximity between newly synthesised callose and cellulose microfibrils and 48 h induction times produce phenotypes that resemble those reported in plants defective in cellulose organisation and cell wall crystallinity[56] (Fig. 8 and Supplementary Fig. 10). These results support the potential applicability of our hydrogel model in the understanding callose-cellulose interactions in vivo.

Traditionally, callose production has been associated with cell wall reinforcement to restrict the penetration of pathogens or in response to elicitors and other stress conditions[18,27]. Our results using hydrogel model suggest that increasing callose concentration significantly reduces the Young's modulus of neat cellulose by about three fold, increasing flexibility, whilst not affecting the viscous nature (plasticity values), hence its ability to absorb deformation energy, until quite high concentrations. It also changes the material properties beyond the yield point, acting as a plasticizer and preventing the catastrophic failure of the material under a high strain condition. Adding callose appears to increase cellulose resilience to high strain through a plastic deformation after the yield point rather than the total failure and fracture that occurs in a more crystalline material. Based on this data, we speculate that callose enhances cell walls resilience in high strain conditions by increasing flexibility preventing the build-up of high stress, and by allowing a failure mode that involves a gradual deformation rather than a catastrophic tearing apart. Toughness, in mechanical engineering terms, is the ability to deform without breaking, and at the same time absorbing and dissipating the energy, which requires a balance of deformability and energy absorption via viscous damping or plastic deformation/ductility[57]. Whilst the addition of callose does not increase toughness in strict terms (the area under the stress-strain curve to the yield point is not increased) it appears to increase ductility under high strain condition. Cell wall mechanics is tightly dependent on the cell wall environment and, based on our research in hydrogel models, callose has the capacity to increase the resilience of cellulosic materials to large deformations. The importance of these features of callose in the regulation of cell wall microdomains (such as plasmodesmata) is difficult to predict without more knowledge on the structural architecture and mechanical properties of these cell walls domains. The complexities of this challenge increase when considering cell wall metabolic cross-talks and dynamic response to signals.

Our results challenge models where callose is viewed as a carrier of cell wall stiffness and support the establishment of interactions between callose, cellulose (and potentially other polymers) as determinant of cell walls responses to developmental

and environmental signals. It also highlights the importance of considering cell wall composition and dynamics and how these affect callose biological function. More studies are required to dissect callose properties and potential role as cellulose modifier in cell walls. From the material science perspective, the interactions between callose and cellulose, exposed by this research, open the path for the development of new applications for this natural biopolymer to create, for example, composites that can be used as biodegradable substitutes for other less environmentally friendly polymers.

## Methods

**Materials**. Microcrystalline cellulose Avicel PH-101, the ionic liquid 1-ethyl-3-methyl-imidazolium acetate ([C2mim][OAc]) (97% purity), FTIR grade KBr, ethanol (absolute, ≥99.8%), acetone (≥99.9%), methanol (≥99.9%) and chloroform (≥99%) were all obtained from Sigma Aldrich. Callose commercial analogue, the (1–3)-β-D-glucan Pachyman and aniline blue fluorochrome were obtained from Biosupplies Australia (www.biosupplies.com.au). The molecular weight of Avicel PH-101 is 28 400 g mol$^{-1}$[158] whereas of Pachyman it is as 168,000 g mol$^{-1}$[159]. The molecular weight distribution of Avicel PH-101, as well as polydispersity and crystallinity index were reported elsewhere[60,61]. Molecular weight and other properties (such as polydispersity, flexibility and conformation of polymers) were also reported elsewhere for Pachyman[62].

All plant work used *Arabidopsis thaliana* in the Columbia (Col-0) background. Transgenic p*G1090::icals3m* was generated by Agrobacterium-mediated transformation of Arabidopsis plants with a vector containing a hyperactive callose synthase mutant version (*cals3m*) under the control of a β-estradiol inducible p*G1090* promoter[50]. In this transgenic line, callose amount is significantly higher 6–24 h after exposure to β-estradiol in comparison to wild type subjected to the same conditions. Wild type and transgenic seeds were surfaced sterilised in bleach solution (20% thin bleach, 0.01% tween-20) for 10 min and stratified in the dark at 4 °C for 5 days. Seeds were germinated on ½ Murashige and Skoog (MS) medium plates containing 0.22% Murashige and Skoog basal medium (Sigma Aldrich), 1% sucrose (Sigma Aldrich), 0.8% plant agar (Sigma Aldrich) and grown under long day conditions (16 h photoperiod) at a constant temperature of 20 °C. Ten day old p*G1090::icals3m* seedlings were transferred for induction to ½ MS plates supplemented with 10 μM β-estradiol (Sigma Aldrich, diluted from a stock solution in DMSO). Control experiments (with basal/non-induced amount of callose) were performed by exposing p*G1090::icals3m* to the corresponding solvent (DMSO) concentration or by transferring wild type seedlings to 10 μM β-estradiol. Plants were analysed or harvested at different times after transfer as indicated in Results. For FTIR analysis, plants were collected 24 h post-transfer and flash-frozen in liquid nitrogen to stop metabolic activity before undergoing freeze drying.

**Preparation of ionic liquid mixtures**. Callose was milled at 50 Hz for 2 min in a TissueLyser LT (Qiagen, Hilden, Germany) to break agglomerations formed due to ambient humidity. Cellulose and callose were then dried in a vacuum oven at 50 °C for 2 days prior to preparing the mixes. Five different samples were prepared with concentrations of 0, 20, 50, 80 and 100% callose, all at a final concentration of 10 wt % of total carbohydrate weight in [C2mim][OAc]. First, dry polymers were mixed in the proportions mentioned above and then the mixed powders were dissolved in ionic liquid. All solutions were prepared in an MBraun Labmaster 130 atmospheric chamber under nitrogen, providing a dry environment, with the chamber being maintained at a dew point level between −70 and −40 °C. The [C2mim][OAc] and callose/cellulose were mixed and kept on magnetic stirrer at 50 °C until producing clear solutions. All samples were kept sealed when not in use to prevent any moisture contamination.

**Hydrogel preparation and callose quantification**. The mixtures were hydrated to form hydrogels by adding deionised water, which was renewed every few hours in the course of two days until full removal of the ionic liquid.

The percentage of callose in the hydrogels was determined using aniline blue. The fluorochrome produces a signal directly proportional to (1,3)-β-glucan concentration that can be measured using a fluorescence microscope and/or fluorometer. For confocal imaging, hydrogel samples were cut into 3 × 3 × 1 mm sections and incubated for 30 min at room temperature in 1 ml of staining solution containing 25 mg ml$^{-1}$ of aniline blue fluorochrome in distilled water. Stained samples were washed with 3 changes of distilled water (5 min per change) and mounted on microscope slides. Samples were imaged on a Zeiss LSM880 Upright confocal microscope (Zeiss, Cambridge, UK) using a ×40 objective with 405 nm excitation and 450 nm emission. Imaging settings (pinhole, gain, etc.) were kept constant in all images.

For fluorometric quantification, hydrogels were freeze dried, then ground using a TissueLyser LT (Qiagen, Hilden, Germany). 0.2 mg of the powder was incubated in 200 μl of 1 M NaOH at 80 °C for 30 min then centrifuged at 11,336×*g* for 15 min to remove any undissolved material. The supernatant was added to 1.25 ml of aniline blue staining solution (0.25 M glycine, pH 9 with 2.25% aniline blue) and

incubated for 20 min at 50 °C followed by 30 min at room temperature. 200 μl of sample was aliquoted to a dark multiwell plate and fluorescence was assayed at 405 nm excitation, 512 nm emission using a POLARstar OPTIMA microplate reader (BMG labtech, Ortenberg, Germany).

A calibration curve was created using different starting weights (0.05, 0.1, 0.15 and 0.25 mg) of callose (powder obtained from the 100% callose hydrogel). Changes in fluorescence values in relation to callose weight (mg) were represented to generate a linear regression curve. This was used to calculate callose content in each of the hydrogels as described above.

**Characterisation of hydrogels using SEM**. Hydrogels were fixed for 2 h in 2.5% glutaraldehyde in 0.1 M phosphate buffer, then washed twice in 0.1 M phosphate buffer for 30 min each. They were then incubated with 1% osmium tetroxide in 0.1 M phosphate buffer overnight. Gels were dehydrated using an ascending acetone series (20–40–60–80–100%) for 30 min each change, followed by critical point drying with a Polaron E3000 apparatus using carbon dioxide.

The specimens were then mounted on 13 mm diameter pin stubs using double sided adhesive carbon tape tabs. These were then coated with platinum to a thickness of 5 nm using a Cressington 208 h high resolution sputter coating unit. Images were then obtained using a FEI Quanta 200 F FEGESEM (field emission gun environmental scanning electron microscope).

**AFM-nanoindentation analysis**. AFM coupled with a colloidal probe was used to probe the mechanical properties of the hydrogels. A few drops of cellulose-callose mixtures in ionic liquid were placed in an O-ring fixed on a glass slide, and then the whole slide was immersed in a petri dish with methanol for 5 h with continuous exchange of methanol every 30 min. When the alcogel disc was formed, it was then placed in an eppendorf tube filled with deionised water to exchange the methanol for water. The samples were stored in water for a minimum of one day, and usually two days, to ensure complete exchange. This was important as even residual swelling at rates of nanometres per second would take the sample surface beyond the limited z-range of the AFM scanners with the long acquisition times of the force spectroscopy maps. In preparation for the AFM, the hydrogels were briefly placed onto a filter paper to remove excess water on the underside only and then glued on a glass slide using two-part epoxy.

An Asylum Research MFP-3D AFM (Asylum Research, Santa Barbara, CA, USA) and a PNP-TR silicon nitride cantilever with polystyrene colloidal probe (SQube, Surface Science Support, Germany) with diameter of 1.98 μm was used for measuring the elastic modulus of the hydrogels. The cantilever was calibrated for sensitivity and spring constant before starting any measurement. The spring constant was calibrated in water using the thermal tuning method[63] giving 0.09 N m$^{-1}$. The gel disc for each sample was prepared twice and for each sample force volumes were collected on an average of 5 different areas of the gel. An average of 5 Force maps (20 × 20 μm) were acquired for each sample on different areas, with each force volume map providing 400 individual force measurements. The loading velocity was fixed to 1 μm s$^{-1}$, ramp size of 1 μm and a maximum applied load of 20 nN. For all samples, the same probe was used and all the parameters were kept constant. Between experiments the cantilever was rinsed with Isopropanol, then ozone cleaning was performed to ensure the colloidal probe is not contaminated.

The Hertz model was used for calculating a Young's modulus $E$ of the hydrogels. This models a sphere indenting a flat plane, the exact geometry of our system. The force $F$ applied is given by:

$$F = \frac{4E\sqrt{R}}{3(1-\alpha)}\varnothing^{3/2} \qquad (2)$$

where $\Phi$ is the indentation depth of the colloidal probe into the hydrogel, $\alpha$ is the Poisson ratio of the hydrogel (here estimated ≈ 0.35) and $R$ is the radius of the probe sphere. The proprietary AFM (Asylum Research) software was used to fit the Hertz model to the first 20% indentation only (Supplementary Fig. 2, black trace on the force curve around the contact area).

The viscoelastic properties have been calculated from the hysteresis between loading and unloading for all samples, which is related to the dissipation of energy during sample deformation. The plastic behaviour of the hydrogel can be quantified by calculating the plasticity factor ($P$) from the ratio between the areas under the unloading ($A_u$) and loading curves ($A_L$) as given by Eq. 3 which reflects the relative plastic/elastic behaviour of the material under force[40,41,64]

$$P = 1 - \left(\frac{A_u}{A_L}\right) \qquad (3)$$

For purely elastic samples, where the matter recovers immediately and elastically after indentation, the loading and unloading curves overlap and the two areas are equal ($A_U = A_L$), there is no hysteresis and $P = 0$. In contrast, $P = 1$ indicates fully plastic properties where the area under the unloading curve is zero, and the deformation of the hydrogel remains constant when the load is removed, meaning it has been plastically deformed over a timescale of seconds (it might recover at longer time scales).

**Macro-indentation measurement.** Cellulose-callose mixtures in ionic liquid were produced and rehydrated similarly as described above for AFM disc-hydrogels but on a larger scale. 2 ml of cellulose-callose mixtures in ionic liquid were prepared in 5 ml vials and then immersed in methanol for 5 h with continuous exchange of methanol every 30 min. The samples were then incubated in deionized water for 24 h. Hydrogel discs of 10 mm diameter were cut out with a hole borer, the thickness of the gels was approximately 4 mm. 100% callose hydrogels could not be made by the method. The hydrogel discs were glued onto a petri dishes using two-part epoxy and measurements (force-distance) were performed with a TA-TX2 Texture Analyser (stable Micro Systems, Surrey, UK) in compression mode equipped with a flat-bottomed probe of 2.0 mm diameter[65]. Indentation was strain controlled to an indentation depth of around 80% of the full sample depth. The raw force-distance values were exported and analysed. Young's modulus was calculated according to the Sneddon model[66], which is suitable for deep indentation into soft materials that conform to the indenter probe geometry. The Elastic modulus ($E$) was obtained using the following Eq. (4):

$$E = \frac{\sqrt{\pi}}{2} \cdot \frac{S \cdot (1 - \nu^2)}{\sqrt{A}} \tag{4}$$

where $A$ is the contact area, $\nu$ is the Poisson ratio (0.3) and $S$ is the stiffness calculated by $\frac{\delta P}{\delta h}$, where $P$ is the indenter load and $h$ is the displacement of the indenter.

**Rheological measurements.** Rheological measurements of the neat solutions and of cellulose/callose mixtures in [C2mim][OAc] were performed on Kinexus rheometer (Malvern Instrument Ltd., Worcestershire, England, UK), equipped with cone-plate geometry ($4^0$–40 mm) with 2 mm gap and a temperature controlled system. Viscoelastic spectra of the samples (elastic component, $G'$, viscous component, $G''$, and phase angle $\delta$) were detected with the oscillation frequency sweep method, within a frequency range 0.01–100 Hz, at 25 °C. For a pure elastic material (solid-like behaviour) the stress and strain are in phase with each other resulting in $\delta = 0$, but for a pure viscous material (liquid-like behaviour) the stress and strain are out of phase[67] with $\delta = 90^0$.

Steady state viscosity of each sample was measured using the same rheometer at increasing shear rate in the range of 0.01–100 s⁻¹ across the temperature range 30–60 °C, in 10 °C increments.

The activation energy was calculated using the Arrhenius equation (Eq. 5), where $\eta_a$ is the zero shear rate viscosity, $A$ is a constant, $R$ is the universal gas constant (8.314 J mol⁻¹ K⁻¹) and $T$ is temperature in K.

$$\eta_a = A e^{\left(-\frac{E_a}{RT}\right)} \tag{5}$$

**Pulsed-field gradient ¹H NMR spectroscopy.** The cation [C2mim]⁺ and anion [OAc]⁻ self-diffusion coefficients were measured as described elsewhere[45] by pulsed-field gradient ¹H NMR, using an Avance II NMR Spectrometer (Bruker Biospin) with a ¹H resonance frequency of 400 MHz. The measurements were carried out in a Diff50 diffusion probe (Bruker Biospin), which can generate a field gradient of up to 20 T m⁻¹. Before each measurement the calibration of the gradient field strength was confirmed by determining the diffusion coefficient of water at 20.0 ± 0.1 °C, this has the value of $(2.03 \pm 0.01) \times 10^{-9}$ m² s⁻¹. The sample environment temperature was also confirmed, by experimentally determining the temperature dependence of the diffusion coefficient for water and comparing those results to data published by Holz et al.[6] To reduce the effect of convection on our results we followed the advice of Annat et al.[6] by keeping NMR tube sample depths to less than 1 cm. The uncertainty in the obtained diffusion coefficients is estimated to be less than 3%. A stimulated echo pulse sequence with bipolar gradients was used, in which the attenuation of the signal intensity follows the following Eq. 6[70]:

$$\ln\left(\frac{S_i}{S_{i0}}\right) = -D_i \gamma^2 g^2 \delta^2 \left(\Delta - \frac{\delta}{3} - \frac{\tau}{2}\right) \tag{6}$$

where $S_i$ is the measured signal intensity of species i and $D_i$ is the diffusion coefficient of that species, $S_{i0}$ defines the initial signal intensity, $\gamma$ is the proton gyromagnetic ratio, $\delta$ is the pulse duration of a combined pair of bipolar pulses, $\tau$ is the period between bipolar gradients, $\Delta$ is the period separating the beginning of each pulse-pair, and $g$ is the gradient strength. In each experiment the strength of the gradient pulse was incremented, while $\delta$ (2–5 ms), $\Delta$ (60 ms) and $\tau$ (2 ms) were all kept constant. The 90° pulse width was 6.6 μs, $g$ had a maximum value of 600 G cm⁻¹, the number of scans was 16, and the repetition time was 6 s, satisfying the criterion of being at least five times $T_1$, which we measured to be ~1000 ms. Samples were studied in steps of 10 °C over the inclusive temperature range 20−60 °C.

**FTIR spectroscopy.** Hydrogel samples were snap-frozen in liquid nitrogen and then dehydrated in a freeze drier at −80 °C for 24 h. The dried hydrogels were ground using tissue lyser LT (Qiagen, Hilden, Germany) for 20 min each. For the plant samples, alcohol insoluble residues (AIR) were obtained from frozen tissues via extraction in an ascending ethanol series (70, 80, 90, 100% v/v in H₂0) for 1 h per change, followed by extraction with 100% acetone and methanol:chloroform (3:2) (1 h per change). The pellet was recovered by centrifugation between each step. The resulting pellet was dried and used directly for FTIR. 1 mg of each sample was mixed with 300 mg of fresh FTIR grade KBr, ground together, and then compressed to a disc under 10 tons applied weight for 2 min.

FTIR spectra were obtained for all samples using Bruker IFS-66 spectrometer equipped with a liquid nitrogen cooled MCT detector, with all optics under vacuum. Scans were taken at a 2 cm⁻¹ resolution, within frequency range of 400–4000 cm⁻¹, in absorption mode, and 1000 spectra were co-added to yield spectra of high signal-to-noise ratio. Obtained spectra were normalised to the absorbance of the O–H in-plain deformation band at 1336 cm⁻¹.

The energy of the hydrogen bond ($E_H$) was calculated using the Eq. 7[48]:

$$E_H = \left(\frac{1}{K}\right) \left[\frac{(\nu_0 - \nu)}{\nu_0}\right] \tag{7}$$

where $\nu_0$ is the standard frequency corresponding to free –OH groups (3600 cm⁻¹), $\nu$ is the frequency of the bonded –OH groups and $K = 1.68 \times 10^{-2}$ kcal⁻¹.

Lateral Order Index (LOI) was calculated as a ratio of intensity of the peaks, $\alpha_{1429/893}$[71].

**Analysis and confocal imaging of the plant material.** Plants were grown as indicated in the methods section. For phenotyping, seedlings were stained in a two-step procedure by aniline blue followed with Direct Red 23 (Sigma Aldrich, USA). Roots were dissected from the plant hypocotyls and incubated for 2 h at room temperature in a 25 μg ml⁻¹ (42.66 μM) solution of aniline blue fluorochrome (Biosupplies, AU) in a 50 mM K₂HPO₄ buffer under vacuum (60 MPa). A 1% wt. filtered (0.45 μm filter) stock solution of Direct Red 23 (Sigma Aldrich, USA) was then added to the aniline blue solution in order to reach a 0.1% wt. final concentration, and allow to stain one more hour in the same vacuum conditions (60 MPa). Slight changes were made to perform this double staining in leaves: leaves were dehydrated in 96% vol EtOH until cleared, quickly rehydrated in water prior to a 4 h incubation at room temperature in a 25 μg ml⁻¹ (42.66 μM) solution of aniline blue fluorochrome in a 50 mM K₃PO₄ buffer under vacuum (60 MPa). Direct Red 23 was then added as described above to the aniline blue solution and allowed to stain 2 h more. Roots and leaves were then mounted in a 1:1 vol solution of AF1 antifadent (Citifluor, USA) and 50 mM K₂HPO₄ or K₃PO₄ buffer, and subsequently imaged by confocal laser scanning microscopy (Zeiss LSM700). Aniline blue staining was imaged with a 405 nm solid state laser, whose emission was recovered with a short pass 490 nm filter, and Direct Red 23 staining was imaged with a 555 nm solid state laser, whose emission was recovered with a long pass 560 nm filter. Pictures were processed using the deconvolution software Huygens and ImageJ 1.51 s (for z-stacks projections and contrast). Adobe Photoshop CS5 was used to create the montages.

Callose amounts were quantified in AIR extracts obtained 24 h after estradiol treatment using aniline blue fluorescence as described for the hydrogels above.

**Materials and correspondence.** Correspondence to Y.B.-A. or M.R.

## Data availability

All raw and processed data, associated with both main and supplementary figures and tables, are accessible in 'https://doi.org/10.5518/225'. All other data supporting the findings of this study are available within the manuscript or are available from the corresponding authors upon request.

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

## Acknowledgements

This work was supported by EPSRC Grant EF/M027740/1 (which funded M.C.H.-G. work) and the Leverhulme Trust Grant RPG-2016-136 (which funded R.H.A.-S., C.P. and S.A. work). We thank Prof. Peter Hine, Prof. Richard Bushby and Dr. Richard Blackburn for help with the analysis of the data and comments on the manuscript. M.E. R. is grateful to the Royal Society for funding his Industrial Fellowship. Y.H. has been supported by the Finnish Centre of Excellence in Molecular Biology of Primary Producers (Academy of Finland CoE program 2014–2019) decision #271832. Y.H. laboratory was funded by the Gatsby Foundation [GAT3395/PR3]; the National Science Foundation Biotechnology and Biological Sciences Research Council grant [BB/N013158/1]; University of Helsinki [award 799992091], the European Research Council Advanced Investigator Grant SYMDEV [No. 323052].

## Author contributions

All authors contributed to manuscript preparation. R.H.A.-S. and M.C.H.-G. did most of the experiments and contributed to data analysis and interpretation of results. S.A. experimentally determined (1,3)-β-glucan concentration, did all plant work and isolated cell walls extracts for FTIR. C.P. did the macro-indentation measurements. M.F. did the SEM experiments and T.B. contributed to the analysis and interpretation of the data. M. B., S.M. and Y.H. obtained and analysed the *pG1090::icals3m* lines and contributed to the writing of this manuscript. Y.B.-A., M.E.R. and S.D.C. conceived and designed research, interpreted the data and supervised the experiments.

## Additional information

**Competing interests:** The authors declare no competing interests.

