## [Peer Review File · Nature Communications]

Reviewers' comments:

Reviewer #1 (Remarks to the Author):

Abou-Saleh and coworkers have submitted a manuscript on the interactions between callose and cellulose model systems in ionic liquid solutions and hydrogels. The paper deals with a very interesting topic aiming at gaining insight into the specific interactions between cellulose and callose. However, from my point of view the manuscript is a bit speculative in terms of the conclusions drawn from the analysis:

The title is partly misleading, since there is almost no insight gained about the interaction of both biomacromolecules in the cell wall. Generally, it is a difficult task to draw conclusions from the analysis of the interplay of model substances (including pretreatment steps) on the molecular interactions in native cell walls.

The authors base their interpretation of specific interactions on the deviation of the behavior from classical rule of mixtures. However, I think this is rather critical, because the studied polymer mixtures do as well deviate from the ideal model systems. For instance, the theoretical upper bound for axial loading and the lower bound for transverse loading refer to a fibre composite, which consists of continuous and unidirectional fibres. In case of the observed hydrogels neither the type of composite structure nor the loading condition fulfil these basic assumptions.

The authors present a plasticity index in Fig 2b, which is derived from the loading/unloading curves. However, the curves in Fig 2a show a change in indentation modulus and in the viscoelastic behavior depending on the callose content, but no plastic behavior.

The authors assume hydrophobic interactions between cellulose and callose at 20% and 80%. The evidence for this assumption is not clear and it is rather questionable while these hydrophobic interactions should have an impact at 20% and 80%, but not at 50%.

The conclusion that the addition of callose modifies cellulose structure is rather speculative based on the obtained data. The FT-IR data (in particular in Fig. 7) shows a decrease of almost all peaks. XRD measurements should be performed, which can directly prove, whether the crystallinity of cellulose is affected. If so, it would be important and highly interesting to discuss a mechanism that can cause this.

The use of the terminology in terms of mechanical behavior needs to me more specific, e.g. "more elastic (less plastic)".

The statement: "Based on the results, we propose that callose and cellulose engage in hydrophobic in-teractions affecting intramolecular and intermolecular hydrogen bonding...". is too vague (see comment above).

"The effect of callose is comparable to that of a "plasticizer" breaking cellulose-cellulose hydrogen bonds and thus decreasing Young's modulus, crystallinity and hydrogen bond energy." See comments regarding mechanical properties.

The discussion of a potential implementation in engineering materials is too speculative: "...can result in a tough/strong "polymer composite"". Toughness and strength were not investigated in this study.

Reviewer #2 (Remarks to the Author):

The submitted article "Interactions between callose and cellulose in cell walls revealed through the analysis of biopolymer mixtures" deals with the function and the organization of cellulose and calls in plant cell walls.

The effect of callose in cellulose materials on the mechanical properties and the non linearity are new and of interest for the scientific community.

Here are some general and specific comments on the manuscript.

General, Units:

The article is well written and generally easily understandable.

* Yet, there are long sentences that make difficult for the reader to follow all the ideas.

For example, in the introduction, the second sentence is 5 lines long, the 4th is 4 lines long. There are many other examples all along the manuscript. The author may work on it to make the reading easier.

* There are some typing mistakes:

1) Results and discussion, first part, 2nd paragraph, last sentence: "seems to lead to a more fine morphology" should be replaced by "seems to lead to a finer morphology".

2) To the best of my knowledge, one should write about K and not °K (material and method, rheology, explanation of equation 4).

3) the author may also do a quick check of the punctuation. Some is missing.

* There are also some inhomogeneities in the units. More specifically, °C is not written the same through all manuscript. The author may correct it in the text and in the figures.

* The authors may also try to improve the quality of figure 5, 6 and S2.

Figure S2, it would be interesting to get an insert of the region at 0 nm distance. This is the prominent part of the curve, where the indentation is visible and where one can see if the fitting is good or not. That would help the reader to understand.

There are also some questions about the results.

1) I would start with a general comments coming from the "material and method" part. The particles used for this study are commercially available cellulose and callose. Due to the molecular weight given, it seems that the callose is much longer than cellulose. It would be interesting to characterize the used molecules to get an idea about the length, the crystallinity and the surface charge density. All these parameters have huge effects on the molecules interactions and could have an impact on the results.

More over the molecules are refined. What is the effect on the starting materials?

The author are claiming at H bond interactions and Van der Waals interactions. What about the amorphous parts and entanglement?

The flexibility of the particles may have a specific effect on the rheology, AFM, H NMR and FTIR results.

2) For AFM measurements, the author say that they used the same tip for all the experiments. Is it possible to give some information about how they check for possible contamination?

How many experiments were done for each point? to get an idea of the reproducibility.

3) For rheology, Is it possible to have any indication about the reproducibility of the measurements?

If few experiments were done on rheology, can the author add an estimation of the dispersion on the curve or in the text?

To me the first point of the 50% callose/cellulose mixture may be discarded if not reproducible.

Then, as highlighted with complex moduli, 50% and 80% are almost overlapping. So I'm not sure that the limit given between 50% and 80% is the best.

4) For H-NMR, Is it possible to get the reproducibility as well.

In conclusion, the submitted article is new, give some information that are of interest for the cellulose/callose/cell wall structure communities. Some parts need some clarifications about the interpretation, the reproducibility of the experiments. Yet, it should be published after the revisions.

Response to reviewers' comments on the manuscript NCOMMS-17-22637

We thank the reviewers for their comments on our paper. We have answered all their questions and followed their suggestions, with these greatly improving the quality of our article. An additional version with all the modifications has also been submitted.

Reviewers' comments are presented in italics and our answers indicated by >>>>

Reviewer #1 (Remarks to the Author):

Abou-Saleh and coworkers have submitted a manuscript on the interactions between callose and cellulose model systems in ionic liquid solutions and hydrogels. The paper deals with a very interesting topic aiming at gaining insight into the specific interactions between cellulose and callose. However, from my point of view the manuscript is a bit speculative in terms of the conclusions drawn from the analysis:

1-The title is partly misleading, since there is almost no insight gained about the interaction of both biomacromolecules in the cell wall. Generally, it is a difficult task to draw conclusions from the analysis of the interplay of model substances (including pretreatment steps) on the molecular interactions in native cell walls.

>>>> We considered the comments from this reviewer and strengthen the section related to cell wall characterization and analysis (read below). We agree with the reviewer that analysing interactions in native cell wall conditions is difficult which highlight the relevance of using model systems. Since most of the assays were done in composite mixtures and interactions cannot be demonstrated in complex cell walls, we changed the title which now reads 'Interactions between callose and cellulose revealed through the analysis of biopolymer mixtures'

2-The authors base their interpretation of specific interactions on the deviation of the behavior from classical rule of mixtures. However, I think this is rather critical, because the studied polymer mixtures do as well deviate from the ideal model systems. For instance, the theoretical upper bound for axial loading and the lower bound for transverse loading refer to a fibre composite, which consists of continuous and unidirectional fibres. In case of the

observed hydrogels neither the type of composite structure nor the loading condition fulfil these basic assumptions.

>>> Indeed these models, as well as most other models for predicting the mechanical properties of a heterogeneous material, are developed for fibre composites but in reality they merely model two phases with different mechanical properties that have been combined in two idealised ways: one to maximise the stiffness by having the stiffer phase spanning the sample in the direction of strain, and the other to minimise the stiffness by having the stiffer phase in layers orthogonal to the strain direction. There are other models for upper and lower bounds, such as Hashin and Shtrikman, but these are less extreme and therefore within the Voigt and Reuss bounds (see <http://silver.neep.wisc.edu/~lakes/VECmp.html>). We do not believe our system has separated into two distinct phases, but even if it had it would not theoretically be able to go outside these two idealised upper and lower bounds; without some change in the molecular bonding of the polymers themselves the data should still lie within the Voigt and Reuss limits. Our samples do though go outside these limits, and therefore it is reasonable to argue that the two polymers must be interacting with each other, changing blend resultant mechanical properties. The Voigt and Reuss models give us the limits on what physical arrangement can do, but it is not sufficient in our case thus we suggest that interactions between the two polymers must occur changing their organization in the hydrogels. In the revised version we decided to keep a panel in Fig.2, showing the lower and upper bounds because it provides useful information on the potential for interactions between these polymers as described above. We have also modified the text (first section of the results) to explain the significance of the models better.

3-The authors present a plasticity index in Fig 2b, which is derived from the loading/unloading curves. However, the curves in Fig 2a show a change in indentation modulus and in the viscoelastic behavior depending on the callose content, but no plastic behavior.

>>>Hysteresis between loading and unloading was observed for all samples as presented in figure 2, indicating a lag in surface recovery following indentation, and is related to the dissipation of energy during sample deformation (i.e. relative plastic/elastic behaviour). This

is the result of the viscous component of a visco-elastic materials behaviour, and will be loading rate dependent (where the elastic modulus would not be) – viscous damping has a velocity term. However, deriving an absolute viscous component, i.e. loss modulus (G''), whilst possible, is an experimental challenge, and the subject of current research in its own right. The index of plasticity, on the other hand, calculated from the hysteresis was used here to compare the *relative* plastic/elastic behaviour of the sample as described before (for example, Briscoe, Fiori, and Pelillo, **1998**. Nano-indentation of polymeric surfaces. Journal of Physics D:Applied Physics. **31**(19): p. 2395-2405, and Klymenko, Wiltowska-Zuber, Lekka, and Kwiatek, **2009**. Energy dissipation in the AFM elasticity measurements. Acta Physica Polonica A. **115**(2): p. 548-551.)

In AFM indentation curves the index of plasticity can be calculated from the ratio between the areas under the unloading (A_u) and loading curves (A_L) as described in Methods, where 0 is purely elastic and 1 is purely plastic (the surface does not recover following relaxation of the applied force). In our case, plasticity lies between 0.2 and 0.45 representing a mixed viscoelastic behaviour (although the hydrogels are still predominantly elastic in behaviour). We have included new supplementary figures (Supplementary figure 2 and 3) and modified the text (see first section of the results) to clarify what plasticity values represent here and how were calculated.

4-The authors assume hydrophobic interactions between cellulose and callose at 20% and 80%. The evidence for this assumption is not clear and it is rather questionable while these hydrophobic interactions should have an impact at 20% and 80%, but not at 50%.

>>> It is not unusual in blends to find certain ratios that are favoured. For example (J. Phys. Chem. B 2012, 116, 12810–12818) in [C2mim][OAc]-water work it was found that the system “prefers” 3 water molecules per EMIMAc. Our system has anion, cation, callose and cellulose all competing / interacting with each other. We have found a 1 to 4 (20%) and 4 to 1 ratio (80%) of cellulose/callose glucose units to be optimum for slowing down the diffusion of [C2mim][OAc], which indicates there are the highest number of available OH groups at these stoichiometric ratios. Why this should happen at these ratios, we don't know and a computer simulation would be useful to give some insight on this. Non-linear changes in the viscosity of the solutions and the mechanical properties of the hydrogels was observed at the same concentrations (notice reduction in elastic modulus at 20% and in viscoelasticity at

80%) supporting the conclusion that, at these concentrations, interactions between the polymers occur. This is not the only case where these non-linear effects have been described. For example in Gordobil et al., *Carbohydrate Polymers* (2014) 112:56-62 and Sundberg et al., *Cellulose* (2015) 22:1943–1953, the mechanical properties of cellulose-xylan films do not vary linearly with the concentration (the Young modulus being higher at 5% in comparison to both 0% and 20% films). Interactions between these polymers have been further verified using solid-state NMR (Simmons et al., *Nature Comm.* 2016).

We present some of these ideas and hypothesis to explain the evidence in the discussion section.

5-The conclusion that the addition of callose modifies cellulose structure is rather speculative based on the obtained data. The FT-IR data (in particular in Fig. 7) shows a decrease of almost all peaks. XRD measurements should be performed, which can directly prove, whether the crystallinity of cellulose is affected. If so, it would be important and highly interesting to discuss a mechanism that can cause this.

>>>Different methods can be used to measure cellulose crystallinity, such as XRD, FTIR, NMR and also DSC. It is accepted that each method has pros and cons and provide different crystallinity values. FT-IR has been used in other publications such as 'Crystallinity changes observed using CBM probes to label sections of cell walls for crystalline cellulose (CBM2a, CBM3a) and amorphous cellulose (CBM4-1, CBM17) displayed close agreement with changes in crystallinity observed with ATR-FTIR techniques' (*Biomacromolecules*, **2011**, *12* (11), pp 4121–4126) and, according to the authors, these appear over-estimated by X-ray diffraction techniques. We have repeated the FT-IR determinations in more than three biological replicas with similar outcomes. To consider reviewer concerns we have modified the example shown in Figure 8 to demonstrate that not all the peaks are reduced. In any case, for calculating hydrogen bond energy the peak intensity is not used but instead the peak position. We have also calculated lateral order index (LOI) which some authors refer to as crystallinity index for the cell wall samples. LOI is a measurement of general structural organization and we found lower values in callose spectra reflecting a more disordered structure.

Moreover, we added new data (new collaboration with the Helariutta group in SLCU, Cambridge, which produced and characterized the inducible transgenic line for callose

(icals3m) to establish the links between callose and cell wall structural organization. Specifically, it was found that after 24 h induction of the transgenic lines, callose accumulates in close proximity to cellulose microfibrils integrated into the matrix (a pattern previously observed after fungal infection by Eggert, et al. Scientific Reports, 2014). 48 h callose induction restricts root growth (Supplementary Information, figure 10), a phenotype that resembles *any1*, a mutant with normal cellulose content but altered in cell wall crystallinity (Plant Physiol. 2013 May;162(1):74-85). Due to their contribution to these determinations, three new authors (from the Helariutta group) are added.

Following the reviewer's advice we did several XRD measurements using WAXD in cell walls extracted from 11 days Arabidopsis developing seedlings and under the supervision of Prof Peter Hines (an expert in this technique). The aim was to evaluate crystallinity before (wildtype seedlings) and after callose induction using the icals3m line. The profiles obtained (pictures can be provided on request) didn't show cellulose crystalline peaks clearly enough, these were of very low intensity, appearing mainly amorphous. Please notice that we are working with 11 days developing primary cell walls which organization differs from secondary and highly crystalline cell walls such as those found in wooden trees or cotton. These experiments could probably be successful if more appropriate/different plant material is used but transformation efficiency in these plants is very low and time consuming.

6-The use of the terminology in terms of mechanical behavior needs to be more specific, e.g. "more elastic (less plastic)".

>>> The samples in this study are visco-elastic, displaying a variation in both absolute elastic modulus (Young modulus) and elastic-plastic balance (i.e. visco-elasticity). We do not measure absolute viscous modulus, only a relative value. Answers to question 3 above also clarifies the use of these terms. We have now more carefully expressed all references to mechanical behaviour to make clear which property we are referring to.

7-The statement: "Based on the results, we propose that callose and cellulose engage in hydrophobic interactions affecting intramolecular and intermolecular hydrogen bonding...". is too vague (see comment above).

>>> The text is modified to soften this statement. In reality we don't know what interactions are responsible. We can merely speculate and put forward something plausible. Our purpose

is to report the interesting findings that are happening between the two polymers, and we have quantified the effects of these through our many and varied techniques, across which a consistent picture has then developed.

8-“The effect of callose is comparable to that of a “plasticizer” breaking cellulose-cellulose hydrogen bonds and thus decreasing Young’s modulus, crystallinity and hydrogen bond energy.” See comments regarding mechanical properties.

The discussion of a potential implementation in engineering materials is too speculative: “...can result in a tough/strong “polymer composite””. Toughness and strength were not investigated in this study.

>>> The statements cited here have been modified. We agree that increase in toughness is rather speculation, but it is based on the interpretation of our results, and related to a likely biological function. The strong decrease in stiffness combined with little change (slight increase) in plasticity values (viscoelasticity) on 20%-50% callose would lead to an increased toughness, a well understood concept in materials engineering. A plasticizer will increase fracture toughness and protect against catastrophic brittle failure. However, we acknowledge this is an interpretation of our data, and would require further experiments to verify. We have modified the text to explain more clearly why we think toughness is increased and to clarify the speculative nature of this statement.

Reviewer #2 (Remarks to the Author):

The submitted article "Interactions between callose and cellulose in cell walls revealed through the analysis of biopolymer mixtures" deals with the function and the organization of cellulose and calls in plant cell walls.

The effect of callose in cellulose materials on the mechanical properties and the non linearity are new and of interest for the scientific community.

Here are some general and specific comments on the manuscript.

General, Units:

The article is well written and generally easily understandable.

** Yet, there are long sentences that make difficult for the reader to follow all the ideas.*

For example, in the introduction, the second sentence is 5 lines long, the 4th is 4 lines long.

There are many other examples all along the manuscript. The author may work on it to make the reading easier.

>>> The sentences have been modified in the introduction and units, wording/sentence construction have been revised throughout the text.

** There are some typing mistakes:*

1) Results and discussion, first part, 2nd paragraph, last sentence: "seems to lead to a more fine morphology" should be replaced by "seems to lead to a finer morphology".

2) To the best of my knowledge, one should wright about K and not °K (material and method, rheology, explanation of equation 4.

3) the author may also do a quick check of the ponctuation. Some is missing.

** There are also some inhomogeneities in the units. More specifically, °C is not written the same through all manuscript. The author may correct it in the text and in the figures.*

>>> Thank you for pointing to these mistakes which have now been revised throughout the text.

** The authors may also try to improve the quality of figure 5, 6 and S2.*

Figure S2, it would be interesting to get an insert of the region at 0 nm distance. This is the

prominent part of the curve, where the indentation is visible and where one can see if the fitting is good or not. That would help the reader to understand.

>>> The quality of the figures has been improved. Supplementary figure 2 has been modified to contain such an inset.

There are also some questions about the results.

1) I would start with a general comments coming from the "material and method" part. The particles used for this study are commercially available callose and cellulose. Due to the molecular weight given, it seems that the callose is much longer than cellulose. It would be interesting to characterize the used molecules to get an idea about the length, the crystallinity and the surface charge density. All these parameters have huge effects on the molecules interactions and could have an impact on the results.

>>> Callose (Pachyman) and microcrystalline cellulose (Avicel) used in the work are standard polymers. Previous published studies characterized the features of callose (Pachyman, Biosupplies) and cellulose (Avicel PH-101, Sigma) commercial analogues. Specifically Engel, et al. *Biotechnology for Biofuels* (2012) reported the molecular weight distribution of Avicel PH-101 as well as polydispersity and crystallinity index. Other papers also refer to Avicel properties including Röder et al., *Polymer* 2001 and those reported at <https://www.sigmaaldrich.com/catalog/product/sial/11365?lang=en®ion=GB>. Regarding Pachyman, the information is less extensive but still comprises a number of papers such as Ding et al, *Journal of Macromolecular Science, Part B* (2001) where the molecular weight, polydispersity, flexibility and conformation are determined. These references are now added to Materials section.

We have also carried out our own assays to characterize the initial polymers, such as thermogravimetric analysis (TGA), determination of molecular weight distribution and conformation in solution of callose and cellulose with MALLS/RI in DMAc/LiCl. Since the results agreed with published information, we considered they are not novel enough to report here. We will add these if the reviewer still considers these data are necessary.

We added some text in the material section pointing to the published work.

More over the molecules are refined. What is the effect on the starting materials?

>>>>We apologize to the reviewer because the poor choice of language when referring at the 'refining' of the products. The TissueLyser was used at 500 rpm for 2 min in order to break (or mill) the 'clumps' that were formed due to humidity at RT and that can delay the drying and dissolution in ionic liquid. The molecular size or properties are not changed in these mild conditions. In any case the starter powder was treated the same for all the mixtures so this step could not explain the interactions observed at certain concentrations. The text has been amended to add clarification.

The author are claiming at H bond interactions and Van der Waals interactions. What about the amorphous parts and entanglement? The flexibility of the particles may have a specific effect on the rheology, AFM, H NMR and FTIR results.

>>> We are not sure to have well understood this comment. We don't know the "flexibility" of what "particles" the reviewer refers to in our models? Both polymers were fully dissolved in ionic liquid and were forming homogeneous solutions and hydrogels, there were no particles present. Both polymers are polysaccharides and they are semi-rigid macromolecules. The rheology is not typical for entangled polymer solution. As far as crystallinity is concerned, we are fully aware that when cellulose is dissolved it is changing from cellulose I to cellulose II allomorph and we are thus comparing in hydrogels LOI of "mixed hydrogel" with LOI in pure cellulose hydrogel (cellulose II) and in pure callose hydrogel. In the plant , there is both crystalline and amorphous components and polymer entanglements may appear. Due to the complexity of the plant system, we can only extract information on the effect of increasing callose on the ordering/crystallinity of cell walls. Based on our results using hydrogel models, we hypothesize that this due to interactions between callose and cellulose but as indicated in the discussion (last and before last paragraphs) other factors might be playing a role.

2) For AFM measurements, the author say that they used the same tip for all the experiments. Is it possible to give some information about how they check for possible contamination? How many experiments were done for each point? to get an idea of the reproducibility.

>>>The used cantilever was calibrated for sensitivity and spring constant before starting any measurement. The gel disc for each sample was prepared twice, for each sample force

volumes were collected on an average of 5 different areas of the gel, with each force volume map providing 400 individual force measurements. Between experiments the cantilever was rinsed with Isopropanol, then ozone cleaning was performed to ensure the colloidal probe is clean and not contaminated. The materials and methods have been modified with this text.

3) For rheology, Is it possible to have any indication about the reproducibility of the measurements? If few experiments were done on rheology, can the author add an estimation of the dispersion on the curve or in the text? To me the first point of the 50% callose/cellulose mixture may be discarded if not reproducible. Then, as highlighted with complex moduli, 50% and 80% are almost overlapping. So I'm not sure that the limit given between 50% and 80% is the best.

>>> If I understand correctly, the reviewer is asking if the result on "not obeying the mixing rule" is within the errors or not. Rheology is accurate to a few % error. We have done several measurements for cellulose and cellulose mixes. In general, the variance (the square of the standard deviation) remained low for all viscosity data points. As example we added a graph for 0% callose at 40 °C to estimate the dispersion on the curves (Supplementary Information, Figure 5). We also modified the text discussing the deviations of viscosity from the ideal mixing rule prediction. This result supports our suggestion of changes in the molecular network at these concentrations which also appeared in NMR determinations. Another important message here is the gel-like properties of callose solution.

As the reviewer suggested we eliminated the first points in Fig.3 as at low frequency the data is noisy. For the same reasons, and to avoid any doubts, we removed the discussion on the interception of elastic and viscous moduli as far as for some mixtures the data at low frequencies are noisy.

4) For H-NMR, Is it possible to get the reproducibility as well.

>>> Diffusion coefficients can be measured to well within 3%. We have repeated the experiments and have measurements at different temperatures (as shown in Fig.5b and Supplemental Fig.6), also for the cation we have various resonances that we track, giving several D values to indicate the uncertainty of D for the cation. The average percentage errors were < 2%. Also our 10% cellulose solution data matches well within experimental

uncertainty with previous published results for 10% cellulose solution data (Biomacromolecules, Vol. 11, No. 11, 2010). We have modified the text and figure legend to make this clearer.

In conclusion, the submitted article is new, give some information that are of interest for the cellulose/callose/cell wall structure communities. Some parts need some clarifications about the interpretation, the reproducibility of the experiments. Yet, it should be published after revisions.

>>> Thank you! we really appreciate the feedback!

Reviewers' comments:

Reviewer #1 (Remarks to the Author):

The authors have improved the manuscript and added further important information. However, I am still not convinced by the arguments related to applying the rule of mixtures and the discussion of a toughening effect based on callose.

In the revised version, the authors elaborated on the influence of the stiffer component in the direction of strain and orthogonal to the direction of strain. However, there are further factors that have to be considered, when interpreting the data on the basis of the model assumptions:

- The nature of the fibre composite is very different

- The loading direction in the AFM nanoindentation test is not in plane, but perpendicular to the cellulose-callose composite

- The AFM nanoindentation is a "near-surface" test, while the model refers to the bulk material behavior. What is the indentation depth?

Therefore, I cannot agree with the interpretation of the data and the conclusions of this part of the manuscript (in particular lines 206-214).

In terms of the mechanical properties of the hydrogels the authors still not clearly distinguish between "plastic" and "viscoelastic" behavior. Furthermore, the interpretation regarding a toughness increase remains very speculative and partly misleading. A reduction in stiffness does not necessarily result in a tougher material and the definition of toughness provided in the manuscript does not reflect crucial factors of the behavior of a "natural" fibre composite. Since the toughness was not investigated in the study, I think there is no sufficient basis for the provided discussion.

Reviewer #2 (Remarks to the Author):

The authors have reviewed their first manuscript taking into account the comments of both referees.

The reply to referees comments are convincing and the changes made in the text complete the missing and uncertainty of the previous version.

That is the reason why I accept the revised version without any changes.

In response to the reviewer's comments, I am pleased that reviewer #2 is satisfied with all the work that we put in answering both reviewers in our last submission. As reviewer #2 accepted the paper without any further modifications, we have here answered the reviewer #1 concerns. In responding to reviewer 1, we have carried out new experiments and added new data to support our hypothesis. We have also revised the writing and when necessary clarified the terminology. Please find the answers to specific points raised in blue below.

Reviewer #1

The authors have improved the manuscript and added further important information. However, I am still not convinced by the arguments related to applying the rule of mixtures and the discussion of a toughening effect based on callose.

Thank you for your in-depth review of our manuscript. We (the authors) appreciate the reviewer's concern on the discussion regarding the toughening effect of callose and, in fact, we are grateful as in response we have carried out new experiments to further test callose effects on the mechanical behaviour of the hydrogels (as discussed below). The results from this experiment support our previous ideas, albeit with a more realistic and nuanced outcome. The main result of the paper is (as the title indicates), that callose and cellulose interact in biopolymer mixtures at the molecular level, evident by comparing the properties of the mixtures with predictions generated using the ideal mixing rule, which applies in mixtures of non-interacting polymers as shown by several authors.

In the revised version, the authors elaborated on the influence of the stiffer component in the direction of strain and orthogonal to the direction of strain. However, there are further factors that have to be considered, when interpreting the data on the basis of the model assumptions:

- The nature of the fibre composite is very different
- The loading direction in the AFM nanoindentation test is not in plane, but perpendicular to the cellulose-callose composite
- The AFM nanoindentation is a "near-surface" test, while the model refers to the bulk material behavior. What is the indentation depth?

Therefore, I cannot agree with the interpretation of the data and the conclusions of this part of the manuscript (in particular lines 206-214).

As we understand, the reviewer raises concerns about applying the Reuss and Voigt model (upper and lower bounds for the Young's modulus based on the rule of mixtures). This model, as most other models for predicting the mechanical properties of heterogeneous materials, are developed for fibre composites, but in reality they merely mathematically model two phases with different mechanical properties that have been combined in two idealised ways: one to maximise the stiffness by having the stiffer phase spanning the sample in the direction of strain (Voigt), and the other to minimise the stiffness by having the stiffer phase in layers orthogonal to the strain direction (Reuss). These mathematical models do not infer the use of fibres and we are aware that our hydrogel samples are isotropic (do not have a preferred orientation) as they do not contain fibres. Indeed, our samples are homogeneous and therefore it is not possible to talk about being 'perpendicular to the cellulose-callose composite'. We found that the model is useful, as it provides an idealized upper and lower modulus limit to explain deviations in linear mixing caused by the internal physical arrangement of the samples that might explain our data. The upper bound of the model coincides with values predicted using the ideal mixing rule which explains the mechanical behaviour of mixtures of non-interacting polymers (the properties of mixtures are the weighted average of the individual components' properties). This upper bound describes well, for example, the mechanical properties of various cellulose-reinforced polymer nanocomposites (Lee et al, Composites Science and Technology Volume 105, 10 December 2014, Pages 15-27). Deviations below the Reuss limit mean that the material properties cannot be explained by differences in internal arrangement, be it fibres or other structures, and so must indicate further interactions between the polymers at the molecular level.

In regard to the reviewers concern about AFM-nanoindentation only reporting on the surface or near-surface, rather than bulk material properties, our new experiments using bulk hydrogel material and a Texture Analyzer equipped with a 2mm flat-ended probe (Fig. 3) answer this point. Young's modulus in relation to callose concentration follows an identical pattern to AFM-nanoindentation (Fig.2). Where the nano-indentation is only to a depth of 50-150 nm (depending on modulus), the bulk measurement was taken all the way to sample failure, with a linear elastic region below the yield stress extending in some samples to a depth of 1 mm, 4 orders of magnitude deeper than the AFM measurement. Whilst perfectly replicating the relative reduction in modulus with addition of callose, the absolute values are 50% lower. Considering the previously discussed orders of magnitude difference in measurement scale, this is perhaps not surprising, and might be due to a possible difference in surface and bulk modulus that the reviewer suspected. However, it might also be explained by the difference in indenter geometry, with a flat bottomed cylindrical indenter rather than a colloidal sphere with the AFM, a different contact mechanics model with largely different parameters (Hertz for AFM vs Sneddon for the flat-ended cylindrical probe), or by a low strain vs high strain measurement. With material properties spanning many orders of magnitude, 50% is relatively small. Indentation depth and other features are described in the methods section.

We have modified the text (including formal lines 206-214) to clarify the use and limitations of the Reuss and Voight model for interpreting the data (highlighted) and modified the conclusions in light of the new results (highlighted).

In terms of the mechanical properties of the hydrogels the authors still not clearly distinguish between "plastic" and "viscoelastic" behaviour.

We have clarified our definition for plastic behaviour (deformation in response to applied forces, see line 180) and, to avoid confusion, limited the use of viscoelastic behaviour to describe rheological properties.

Furthermore, the interpretation regarding a toughness increase remains very speculative and partly misleading. A reduction in stiffness does not necessarily result in a tougher material and the definition of toughness provided in the manuscript does not reflect crucial factors of the behaviour of a "natural" fibre composite. Since the toughness was not investigated in the study, I think there is no sufficient basis for the provided discussion.

We agreed that toughness was not investigated experimentally in the study and was only proposed in the discussion as a potential mechanism to explain the behaviour of callose in natural systems in light of the data we had at that point. We have now carried out new experiments to further evaluate the influence of callose in the mechanical properties of bulk "mixed" hydrogels. Based on the new results the emerging picture is not exactly one of increased 'toughness', but a higher tolerance to conditions of high strain. The material upon failure undergoes a process more akin to plastic deformation, rather than a catastrophic failure and tearing apart as with pure cellulose. We have modified the discussion accordingly. Speculatively this could apply to cell walls but so far this is only a hypothesis.

The main message of the manuscript is that callose and cellulose interact forming mixtures with interesting physico-mechanical properties. This is a new finding on the properties of a polymer that plays an essential function in cellulosic cell walls. Our aim in the discussion is to highlight that this new knowledge should be taken into consideration when interpreting the effect of increasing callose content on cell walls structure and mechanical properties (so far unknown) and provide a platform for investigating the potential exploitation of this knowledge in the generation of new materials.

REVIEWERS' COMMENTS:

Reviewer #1 (Remarks to the Author):

I am fine with the response letter and the revision of the manuscript, although I still feel that it is not too meaningful to use a model for composites with unidirectional, continuous fibres for a hydrogel that does not contain fibres. However, the underlying assumptions are now better traceable by the reader and the authors have added a clear statement on this limitation.

Two minor comments:

Page 8, line 227: I would remove “...(force required to fracture the hydrogels)...”, as this addition is misleading.

Page 19, lines 483-484: The authors should rephrase “...resilience to...high stress...”, in view of the fact that yield stress is reduced by 2-3 times.

Response to reviewer comments:

>>> We thank the reviewer for spotting these mistakes which are now modified for the appropriate text in the current version as detailed below.

Reviewer #1 (Remarks to the Author):

I am fine with the response letter and the revision of the manuscript, although I still feel that it is not too meaningful to use a model for composites with unidirectional, continuous fibres for a hydrogel that does not contain fibres. However, the underlying assumptions are now better traceable by the reader and the authors have added a clear statement on this limitation.

Two minor comments:

Page 8, line 227: I would remove “···(force required to fracture the hydrogels)···” , as this addition is misleading.

>>> This has been deleted in this version

Page 19, lines 483-484: The authors should rephrase “···resilience to···high stress···” , in view of the fact that yield stress is reduced by 2-3 times.

>>> This has now been modified as it should refer to strain not stress (see text 'Adding callose appears to increase cellulose resilience to high strain through a plastic deformation after the yield point rather than the total failure and fracture that occurs in a more crystalline material'.)